# Phosphoglycerate mutase regulates Treg differentiation through control of serine synthesis and one-carbon metabolism

Wesley H Godfrey[1], Judy J Lee[1], Shruthi Shanmukha[2], Kaho Cho[1], Xiaojing Deng[1], Chandra Shekar R Ambati[3], Vasanta Putluri[3], Abu Hena Mostafa Kamal[3,4], Paul M Kim[2], Nagireddy Putluri[3,4], Michael D Kornberg[1]*

[1]Department of Neurology, Johns Hopkins University School of Medicine, Baltimore, United States; [2]Department of Psychiatry and Behavioral Sciences, Johns Hopkins University School of Medicine, Baltimore, United States; [3]Advanced Technology Core, Baylor College of Medicine, Houston, United States; [4]Department of Molecular and Cellular Biology, Baylor College of Medicine, Houston, United States

## eLife Assessment

This paper highlights an **important** physiological function of PGAM in the differentiation and suppressive activity of Treg cells by regulating serine synthesis. This role is proposed to intersect with glycolysis and one-carbon metabolism. The study's conclusion is supported by **solid** evidence from in-vitro cellular and in-vivo mouse models.

*For correspondence:
michael.kornberg@jhmi.edu

**Abstract** The differentiation and suppressive functions of regulatory CD4 T cells (Tregs) are supported by a broad array of metabolic changes, providing potential therapeutic targets for immune modulation. In this study, we focused on the regulatory role of glycolytic enzymes in Tregs and identified phosphoglycerate mutase (PGAM) as being differentially overexpressed in Tregs and associated with a highly suppressive phenotype. Pharmacologic or genetic inhibition of PGAM reduced Treg differentiation and suppressive function while reciprocally inducing markers of a pro-inflammatory, T helper 17 (Th17)-like state. The regulatory role of PGAM was dependent on the contribution of 3-phosphoglycerate (3 PG), the PGAM substrate, to de novo serine synthesis. Blocking de novo serine synthesis from 3 PG reversed the effect of PGAM inhibition on Treg polarization, while exogenous serine directly inhibited Treg polarization. Additionally, altering serine levels in vivo with a serine/glycine-free diet increased peripheral Tregs and attenuated autoimmunity in a murine model of multiple sclerosis. Mechanistically, we found that serine limits Treg polarization by contributing to one-carbon metabolism and methylation of Treg-associated genes. Inhibiting one-carbon metabolism increased Treg polarization and suppressive function both in vitro and in vivo in a murine model of autoimmune colitis. Our study identifies a novel physiologic role for PGAM and highlights the metabolic interconnectivity between glycolysis, serine synthesis, one-carbon metabolism, and epigenetic regulation of Treg differentiation and suppressive function.

## Introduction

Upon activation, immune cells undergo drastic metabolic changes that support their differentiation and effector functions. The regulation of immune responses by metabolic pathways, termed 'immunometabolism,' has become a major focus of research with a goal of identifying pathways that can

be therapeutically targeted in conditions ranging from autoimmune and infectious diseases to cancer (*Patel and Powell, 2017*).

T lymphocytes, in particular, have emerged as a central focus of immunometabolic research with far reaching implications in human disease (*Buck et al., 2016*; *Chapman and Chi, 2022*; *Geltink et al., 2018*; *Kelly and O'Neill, 2015*; *Kornberg, 2020*; *O'Neill et al., 2016*). Among the metabolic adaptations associated with T cell activation, glycolytic reprogramming plays a prominent role. Increased glucose uptake and augmented glycolysis are critical for the differentiation and effector functions of inflammatory T cell subsets, including effector CD8 cells and T-helper (Th) 1 and Th17 CD4 cells (*Frauwirth et al., 2002*; *Cham and Gajewski, 2005*; *Cham et al., 2008*; *Wang et al., 2011*; *Chang et al., 2013*; *Macintyre et al., 2014*; *Gerriets et al., 2015*; *Shi et al., 2011*). In contrast, regulatory T cells (Tregs), which serve as the brakes of the immune response by promoting tolerance and suppressing inflammation, have a distinct glycolytic profile (*Shi and Chi, 2019*; *Angelin et al., 2017*). Prior work has shown that initial engagement of glycolysis supports proliferation and migration of Tregs, whereas downregulation of glycolysis and greater engagement of fatty acid oxidation are required for expression of the Treg-specific transcription factor FOXP3 and Treg suppressive functions (*Ho et al., 2015*; *Wang et al., 2020*; *Lim et al., 2021*; *Kishore et al., 2017*; *Michalek et al., 2011*; *Gerriets et al., 2016*). In this regard, forced engagement of glycolysis was found to decrease FOXP3 expression (*Wei et al., 2016*) while inhibition of glycolysis favored Treg differentiation and suppressive functions (*Shi et al., 2011*; *Scherlinger et al., 2022*; *Kornberg et al., 2018*). The relationship between glycolysis and Treg function is not straightforward, however. For instance, in human Tregs induced following weak stimulation of the T cell receptor (TCR), glycolysis supported Treg suppressive function by regulating DNA-binding activity of the glycolytic enzyme enolase-1 (ENO1) (*De Rosa et al., 2015*). Such findings suggest the possibility of a more nuanced role for glycolysis in Tregs, with individual glycolytic enzymes serving unique and perhaps disparate functions.

Here, we report that the glycolytic enzyme phosphoglycerate mutase (PGAM), which converts 3-phosphoglycerate (3 PG) to 2-phosphoglycerate (2 PG), is physiologically upregulated in Tregs and its enzymatic activity supports FOXP3 expression and Treg suppressive function. Pharmacologic or genetic inhibition of PGAM inhibited Treg differentiation and shifted CD4 T cells toward a proinflammatory, Th17-like state. Mechanistically, we found that PGAM regulates Treg differentiation by controlling flux through the de novo serine synthesis pathway. The PGAM substrate 3 PG is also utilized for serine synthesis, and decreased PGAM enzyme activity led to increased serine synthesis through the accumulation of 3 PG. Both newly synthesized and imported extracellular serine inhibited Treg differentiation by contributing to one-carbon metabolism and promoting the methylation, and therefore silencing, of genes associated with Treg function. Manipulation of serine availability and the methylation cycle increased peripheral Tregs and attenuated disease in mouse models of multiple sclerosis and autoimmune colitis. Our finding that PGAM, a key glycolytic enzyme, physiologically supports Treg differentiation further challenges the notion that glycolysis plays a singular, inhibitory role in Tregs and suggests that individual glycolytic enzymes can serve as nodes for the fine-tuning of immune responses. Furthermore, our work identifies novel metabolic targets for the therapeutic manipulation of Treg function in human disease.

## Results

### PGAM regulates Treg differentiation and suppressive function

In order to identify the glycolytic enzymes that play the largest role in Treg differentiation, we analyzed multiple publicly available transcriptomic and proteomic datasets. We first investigated an RNA sequencing (RNA-seq) dataset that compared human naïve CD4 cells cultured under either Th0 or Treg polarizing conditions (*Ullah et al., 2018*). After library normalization, we examined genes included in the 'Glycolysis and Gluconeogenesis' gene ontology (GO) term and found that *PGAM1* was the most differentially expressed glycolytic gene, with upregulation in in vitro-induced Tregs (iTregs) compared to conventional Th0 cells (*Figure 1A*). To determine whether PGAM expression is similarly upregulated in ex vivo-derived Tregs, we examined a single-cell RNA-seq (scRNA-seq) dataset of peripheral blood mononuclear cells (PBMCs) from healthy human controls (*Schafflick et al., 2020*) and compared glycolytic gene expression between Tregs and total CD4 cells. After removing glyceraldehyde-3-phosphate dehydrogenase (*GAPDH*) for scaling purposes, we found that *PGAM1*

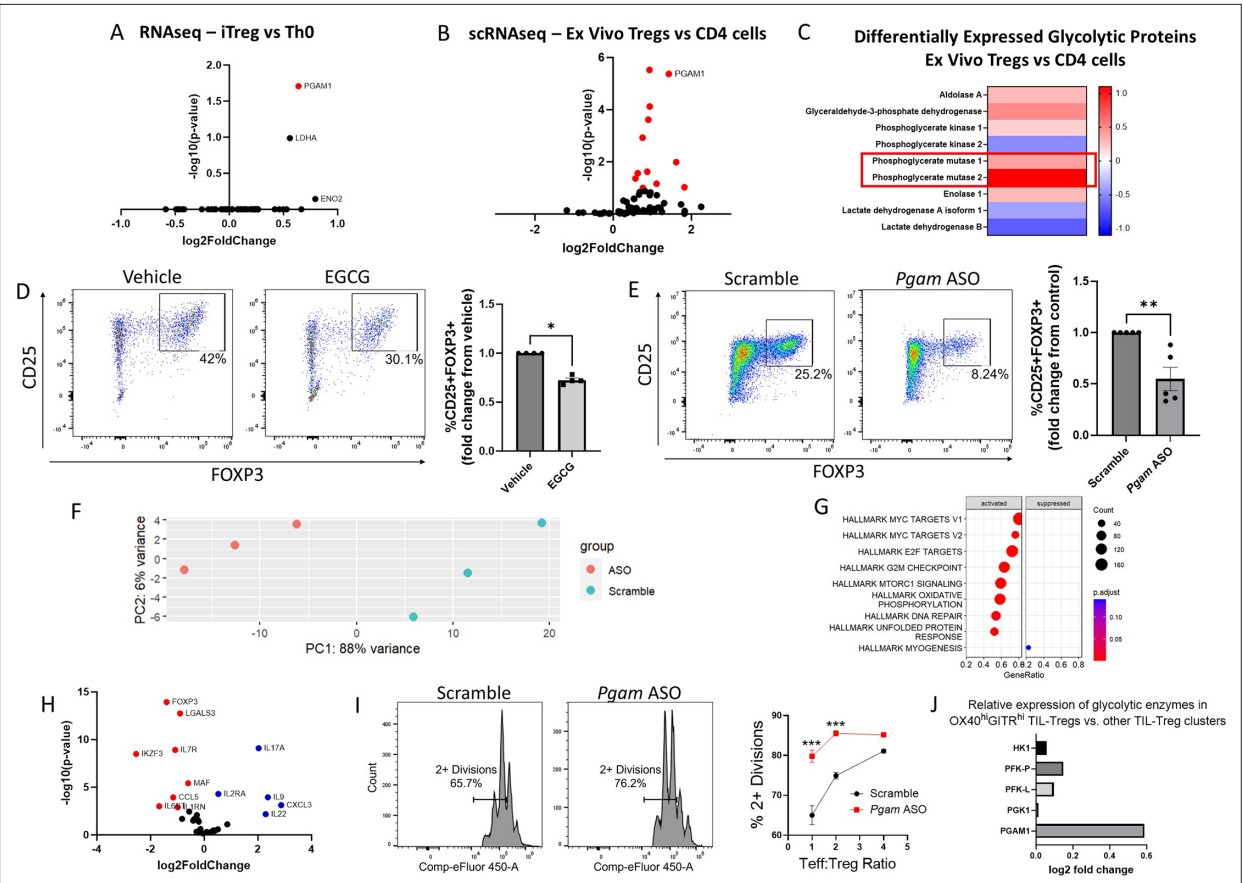

**Figure 1.** Phosphoglycerate mutase (PGAM) regulates Treg differentiation and suppressive function. . (**A** – **C**) Analysis of publicly available transcriptomics and proteomics data reveals upregulation of PGAM expression in human iTregs and ex vivo Tregs. (**A**) Publicly available RNA sequencing (RNA-seq) data from human naïve CD4 cells cultured under either Th0 or Treg polarizing conditions for 72 hr (*Ullah et al., 2018*) was analyzed, and genes belonging to the Gene Ontology (GO) term 'Glycolysis and Gluconeogenesis' were plotted. (**B**) Differential expression of 'Glycolysis and Gluconeogensis' GO genes in ex vivo Tregs vs. total CD4 cells derived from healthy donor peripheral blood mononuclear cells (PBMCs) (*Schafflick et al., 2020*). *GAPDH* was removed for scaling purposes. (**C**) Differential expression of cytosolic glycolytic enzymes in ex vivo Tregs vs. conventional CD4 T cells derived from healthy donor PBMCs, analyzed from publicly available proteomics data (*Procaccini et al., 2016*). (**D**) Naïve murine CD4 cells were cultured under Treg polarizing conditions for 4 days and treated with either EGCG (20 µM) or vehicle on day 1. Treg polarization based on CD25 and FOXP3 expression was analyzed by flow cytometry. Data represent mean ± SEM from four independent experiments, with three to four biological replicates per experiment. (**E**) Naïve murine CD4 cells were treated with either scrambled or *Pgam*-specific antisense oligonucleotides (ASOs) and cultured under Treg polarizing conditions for 72 hr. CD25 and FOXP3 expression were analyzed by flow cytometry. Data represent mean ± SEM from five independent experiments, with three to four biological replicates per experiment. (**F** – **H**) Naïve murine CD4 cells were cultured with either scrambled or anti-*Pgam* ASOs for 72 hr under Treg polarizing conditions. RNA was isolated from unsorted cells for RNA-seq, followed by library size normalization and differential expression analysis. Shown for scrambled versus anti-*Pgam* ASO-treated cells are (**F**) PCA plot, (**G**) Gene Set Enrichment Analysis (GSEA) using MSigDB Hallmark gene sets, and (**H**) volcano plot of genes associated with T cell function. Data from 3 biological replicates. (**I**) Naïve murine CD4 cells were cultured under Treg polarizing conditions with either scrambled or anti-*Pgam* ASOs for 72 hr. To assess Treg suppressive function, the polarized Tregs were cultured with naïve CD4 cells stimulated with CD3/CD28 stimulating antibodies for an additional 72 hr at the indicated ratios. Cell proliferation was measured by dilution of a cell proliferation dye. Data represent mean ± SEM from four biological replicates. (**J**) Analysis of a published scRNA-seq dataset of tumor-infiltrating Tregs (TIL-Tregs) (*Dykema et al., 2023*) shows that *PGAM1* is overexpressed in the most highly suppressive subpopulation, out of proportion to proximal rate-limiting glycolytic enzymes. *p<0.05, **p<0.01, ***p<0.0001 by Mann-Whitney U Test (**D, E**) or two-way ANOVA with multiple comparisons testing (**I**). P-values in (**A**), (**B**), and (**F** – **H**) were derived from Wald's test after False Discovery Rate correction and median-of-ratios normalization via DESeq2.

The online version of this article includes the following figure supplement(s) for figure 1:

**Figure supplement 1.** Epigallocatechin gallate (EGCG) has no effect on Treg viability or proliferation.

**Figure supplement 2.** Efficient uptake of antisense oligonucleotides (ASOs) by cultured CD4 cells.

**Figure supplement 3.** *Pgam* antisense oligonucleotides (ASOs) reduce PGAM1 expression without impacting viability.

**Figure supplement 4.** Glyceraldehyde-3-phosphate dehydrogenase (GAPDH) inhibition promotes Treg polarization.

*Figure 1 continued on next page*

*Figure 1 continued*

**Figure supplement 5.** Increased Phosphoglycerate mutase (PGAM) to glyceraldehyde-3-phosphate dehydrogenase (GAPDH) expression ratio in regulatory T cells.

was similarly upregulated in Tregs (*Figure 1B*). A proteomics dataset from human healthy donor PBMCs *Procaccini et al., 2016* demonstrated significantly upregulated protein levels of PGAM1 and PGAM2 in Tregs compared to total CD4 cells (*Figure 1C*), confirming findings from the mRNA level.

The differential upregulation of PGAM in Tregs suggested that it might play a physiologic regulatory role in these cells. We, therefore, wondered whether PGAM inhibition impacts Treg differentiation. To answer this question, we used epigallocatechin gallate (EGCG), a known inhibitor of PGAM (*Li et al., 2017*). We cultured naïve murine CD4 cells under Treg polarizing conditions with either vehicle (solvent alone) or EGCG and assessed Treg differentiation by quantifying the percentage of CD25$^+$FOXP3$^+$ cells by flow cytometry after 96 hr. Treatment with EGCG significantly reduced Treg differentiation (*Figure 1D*) without affecting proliferation or cell viability (*Figure 1—figure supplement 1*). To validate the effects of pharmacologic inhibition with genetic knockdown, we utilized antisense oligonucleotides (ASOs) targeting *Pgam* isoforms to reduce gene expression. Experiments conducted with fluorescently-tagged ASOs demonstrated near-complete ASO uptake by cultured CD4 cells after 24 hr (*Figure 1—figure supplement 2*). Compared to scrambled control ASOs, we found that ASOs targeting *Pgam* reduced PGAM1 protein levels (*Figure 1—figure supplement 3*) and significantly reduced Treg differentiation assessed by flow cytometry (*Figure 1E*). Because glycolysis is crucial for cell health, we wanted to know if PGAM knockdown was affecting survival and/or proliferation. We found that *Pgam* ASOs did not affect cell survival and led to a minor increase in cell proliferation (*Figure 1—figure supplement 3*).

To gain further insights into the effects of PGAM knockdown on Treg differentiation, we performed RNA-seq transcriptional profiling from unsorted naïve murine CD4 cells cultured under Treg polarizing conditions with either scrambled or *Pgam*-specific ASOs. PCA analysis showed that *Pgam* ASOs led to major transcriptional changes (*Figure 1F*), with major effects on MYC targets and mTORC1 signaling (*Figure 1G*), among other changes. We then performed differential expression analysis on a subset of genes important in the Treg-Th17 axis and found that cells treated with *Pgam* ASOs had significantly higher expression of pro-inflammatory genes such as *Il17*, *Il22*, and *Il9*, while significantly lower expression of genes involved in Treg differentiation such as *Foxp3* and *Izkf3* (*Figure 1H*). A complete list of differentially expressed genes can be found in *Supplementary file 1*.

Given the observed changes in transcriptional profile and differentiation markers, we next asked whether PGAM knockdown leads to functional changes in Treg suppressive activity. Compared to treatment with scrambled ASOs, *Pgam* ASO-treated Tregs were less effective at suppressing effector T (Teff) cell proliferation, indicating diminished suppressive function (*Figure 1I*). To determine whether PGAM expression is associated with Treg suppressive activity in human disease, we examined a published scRNA-seq dataset of tumor-infiltrating Tregs (TIL-Tregs) isolated from surgically resected non-small cell lung cancer (*Dykema et al., 2023*). Among the 10 distinct subsets of TIL-Tregs identified in this study, one subset, the OX40$^{hi}$GITR$^{hi}$ cluster [termed 'Activated (1)' by the authors] had markedly increased suppressive activity and was enriched in patients who were resistant to immune checkpoint blockade. Strikingly, *PGAM1* was significantly overexpressed in the highly suppressive OX40$^{hi}$GITR$^{hi}$ cluster compared to the 9 other TIL-Treg subsets (*Figure 1J*). Moreover, *PGAM1* expression was selectively increased in this subset relative to proximal, rate-limiting glycolytic enzymes, suggesting a unique association between PGAM expression and Treg suppressive activity.

Taken together, these findings demonstrate a critical role for PGAM in promoting Treg differentiation and suppressive function, in vitro as well as in healthy and diseased human states. The positive regulatory role of PGAM in Tregs was intriguing, since glycolysis has generally been associated with decreased Treg function. Indeed, whereas inhibition or knockdown of PGAM led to decreased Treg differentiation, our own prior work found that similar inhibition of GAPDH, which catalyzes a reaction only two steps upstream from PGAM in glycolysis, had the opposite effect (*Kornberg et al., 2018*) – a finding we replicated here with the GAPDH inhibitor koningic acid (*Figure 1—figure supplement 4*). Consistent with this observation, the ratio of *PGAM1* to *GAPDH* mRNA levels increased in human iTregs vs. Th0 cells (*Figure 1—figure supplement 5*), and PGAM2 protein levels increased out of proportion to GAPDH (and other glycolytic enzymes) in ex vivo

Tregs (*Figure 1C*). Similarly, a review of published data from the ImmPres immunological proteome resource (http://immpres.co.uk/; *Brenes et al., 2023*) shows increased PGAM1/GAPDH protein copy number ratio in murine iTregs versus Th17 cells (*Figure 1—figure supplement 5*), further implicating a specific role for PGAM in the Treg-Th17 axis. The discordant roles of GAPDH and PGAM in Tregs led us to focus on the glycolytic metabolites downstream of GAPDH and upstream of PGAM, as the concentrations of these metabolites would be differentially impacted by the two enzymes (*Figure 2A*).

## PGAM regulation of Treg differentiation is mediated by 3PG and de novo serine synthesis

PGAM metabolizes the glycolytic intermediate 3 PG into 2 PG (*DiMauro et al., 1981*; *Koo and Oskarsson, 2016*). In addition to its role in glycolysis, 3 PG is the starting substrate for the de novo serine synthesis pathway (*Lunt and Vander Heiden, 2011*; *He et al., 2023*). Given previous work implicating various roles for serine in T cells (*Ron-Harel et al., 2016*; *Ma et al., 2017*; *Kurniawan et al., 2020*), we hypothesized that PGAM enzymatic activity regulates Treg differentiation and function by controlling levels of 3 PG with downstream effects on serine synthesis (*Figure 2A*). To test this hypothesis, we first cultured naïve murine CD4 cells under Treg polarizing conditions with either vehicle or the PGAM inhibitor EGCG and measured intracellular serine levels from unsorted cells by mass spectrometry. As anticipated, PGAM inhibition with EGCG increased intracellular levels of serine in Tregs (*Figure 2B*). We next asked whether glucose-derived de novo serine synthesis is physiologically regulated during Treg differentiation, focusing on the reciprocal Treg-Th17 axis. Following incubation with uniformly 13-carbon labeled glucose (U$^{13}$C-glucose), serine that is newly synthesized from 3 PG has a mass shift of 3 (M+3) when analyzed by mass spectrometry, whereas unlabeled, non-glucose-derived serine has no mass shift (M+0). We differentiated naïve murine CD4 cells into either Tregs or Th17 cells for 72 hr. After 6 hr glucose starvation, the cells were incubated with U$^{13}$C-glucose for an additional 6 hr. Using mass spectrometry, we found that $^{13}$C-labeling of serine was significantly higher in Th17 cells compared to Tregs (*Figure 2C*), indicating that flux through de novo serine synthesis is physiologically downregulated in Tregs compared to Th17 cells. Because serine can be recycled via one-carbon metabolism, we observed an increase in one-carbon-labeled (M+1) serine as well as M+3 serine in Th17 cells relative to Treg cells.

If the role of PGAM in Treg differentiation depends on de novo serine synthesis, then the effects of PGAM inhibition should be rescued by simultaneously blocking the serine synthesis pathway. To this end, we cultured naïve murine CD4 cells under Treg polarizing conditions and treated them with the PGAM inhibitor EGCG ±NCT-503, an inhibitor of phosphoglycerate dehydrogenase (PHGDH, the rate-limiting enzyme of serine synthesis from 3 PG) (*Bojkova et al., 2023*; *Pacold et al., 2016*). As before, PGAM inhibition with EGCG inhibited Treg differentiation as assessed by FOXP3 expression, but this was reversed by PHGDH inhibition with NCT-503 (*Figure 2D*). Similarly, if PGAM regulates Tregs by controlling concentrations of 3 PG (the substrate for serine synthesis), then direct supplementation of 3 PG should replicate the effects of PGAM inhibition. To test this, 3 PG was delivered to naïve murine CD4 cells via electroporation, followed by Treg polarization. As expected, 3 PG supplementation decreased FOXP3 expression, which was rescued by PHGDH inhibition with NCT-503 (*Figure 2E*). To further validate the role of de novo serine synthesis in Tregs, we used *Phgdh*-targeted ASOs and found that genetic knockdown of PHGDH led to augmented Treg differentiation as assessed by FOXP3 expression (*Figure 2F*).

The above findings demonstrate that PGAM regulates Treg differentiation by controlling flux through the serine synthesis pathway. Because serine can be derived from both de novo intracellular synthesis and extracellular sources, we next asked whether extracellular serine also affects Treg differentiation. We cultured naïve murine CD4 cells under Treg polarizing conditions in either serine/glycine-free media or serine/glycine-free media supplemented with 4 mM serine. Deprivation of both serine and glycine is required to evaluate the role of extracellular serine, as glycine can be converted to serine via the enzyme serine hydroxymethyltransferase (SHMT) (*He et al., 2023*). The presence of extracellular serine diminished Treg differentiation (*Figure 2G*), indicating that both synthesized and exogenous serine impact Treg generation.

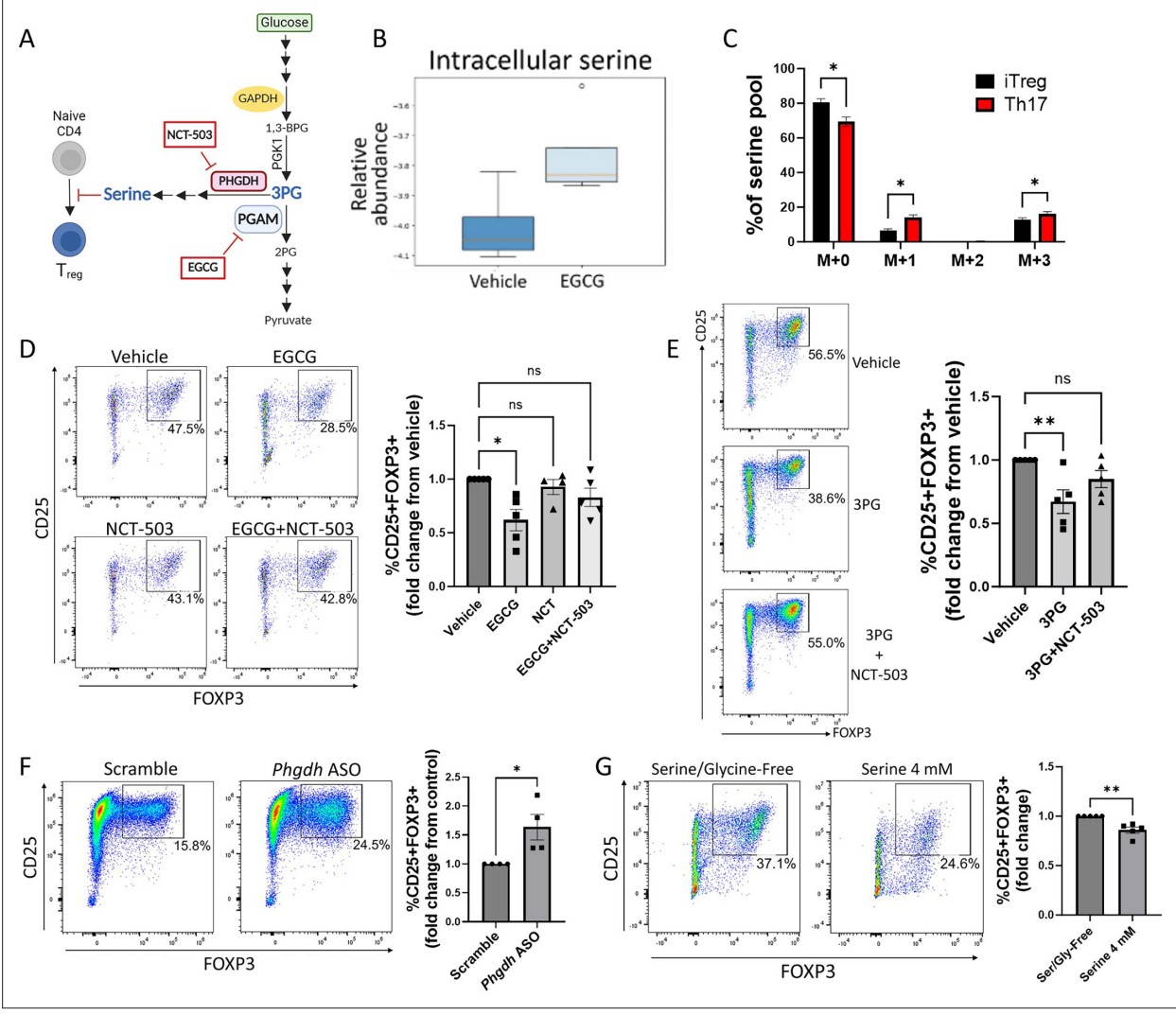

**Figure 2.** Phosphoglycerate mutase (PGAM) regulates Treg differentiation through control of de novo serine synthesis. (**A**) Schematic diagram of the intersection of glycolysis and the de novo serine synthesis pathway via the PGAM substrate 3 PG. Pharmacologic inhibitors are shown in red boxes. This panel was created using BioRender.com. (**B**) Naïve murine CD4 cells were cultured under Treg polarizing conditions for 4 days and treated with vehicle or epigallocatechin gallate (EGCG) (20 μM) on day 1. Unsorted cells were then lysed and metabolites were extracted and analyzed via LC-MS. Data from four biological replicates. (**C**) The rate of glucose-derived serine synthesis from polarized murine Tregs or Th17 cells was measured by adding U[13]C-glucose to the culture media for 6 hr and quantifying the percentage of labeled/unlabeled serine via LC-MS. The mass isotopomer distribution (MID) of U[13]C-glucose-derived serine and percent contribution to the total serine pool is shown. Data represent mean ± SEM from three biological replicates. (**D**) Naïve murine CD4 cells were cultured under Treg polarizing conditions for 4 days. On day 1, the cells were treated with vehicle or the indicated combinations of the PGAM inhibitor EGCG (10 μM) and the PHGDH inhibitor NCT-503 (10 μM) (doses optimized for combination treatment). Treg polarization was assayed by flow cytometry. Data represent mean ± SEM from five independent experiments. (**E**) Naïve murine CD4 cells were polarized under Treg conditions for 4 days. On day 1 of polarization, cells were either sham-electroporated or supplemented with 3 PG (1.5 mM) by electroporation and treated with vehicle or the PHGDH inhibitor NCT-503 (10 μM). Analysis was performed by flow cytometry. Data represent mean ± SEM from five independent experiments. (**F**) Naïve CD4 cells were cultured under Treg polarizing conditions with either scrambled or anti-*Phgdh* ASOs, and Treg generation was analyzed by flow cytometry. Data represent mean ± SEM from four independent experiments. (**G**) Naïve murine CD4 cells were cultured under Treg polarizing conditions for 4 days with either serine/glycine-free media or serine/glycine-free media supplemented with 4 mM serine, followed by flow cytometric analysis. Data represent mean ± SEM from five independent experiments. *p<0.05, **p<0.01 by Mann-Whitney U test (**F, G**), Kruskal-Wallis test with multiple comparisons testing (**D, E**), or Student's t-test with multiple hypothesis correction (**C**). Each independent experiment shown in (**D-E**) included three to four biological replicates.

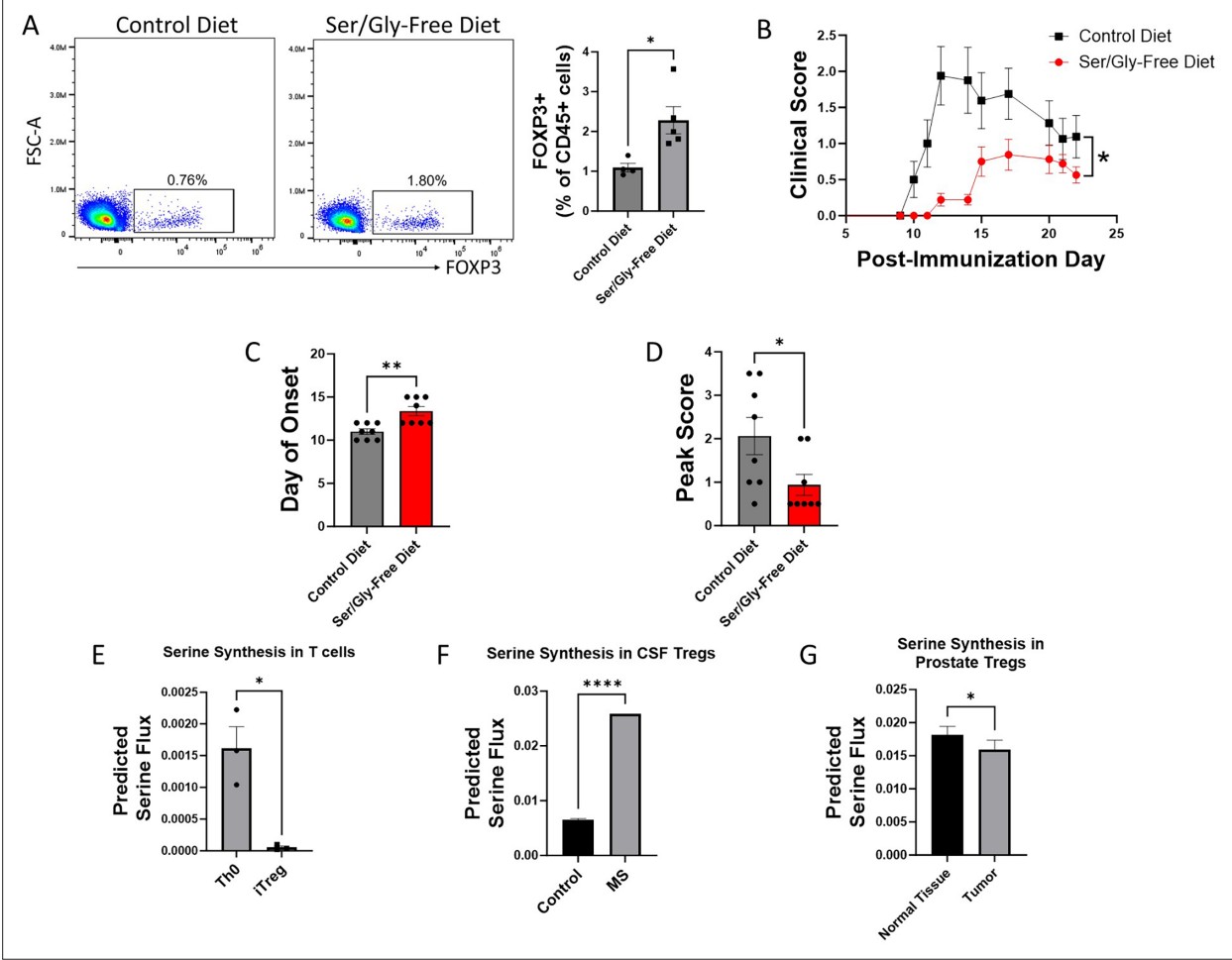

**Figure 3.** Serine synthesis and dietary availability regulate Tregs in vivo and in disease states. (**A**) C57BL/6 mice were fed either serine/glycine-free diet or control diet for 8 weeks, beginning immediately post-weaning. Peripheral Tregs from blood were quantified by flow cytometry as the percentage of CD45[+] cells expressing FOXP3. Data represent mean ± SEM from four to five mice per group. (**B-D**) Mice were fed either serine/glycine-free diet or control diet for 7 days and then subjected to MOG$_{35-55}$ EAE. Clinical scoring was performed by a blinded observer. Data represent mean ± SEM from eight mice per group. Shown are (**B**) clinical scores over the course of the experiment, (**C**) day of neurologic symptom onset, and (**D**) peak clinical scores. (**E-G**) Publicly available RNA-seq data from three different datasets were analyzed using single-cell flux estimation analysis (SCFEA) to model serine synthesis rates from mRNA levels of metabolic enzymes. (**E**) SCFEA from human naïve CD4 cells cultured under either Th0 or Treg polarizing conditions for 72 hr (*Ullah et al., 2018*) predicts decreased serine synthesis in iTregs. (**F**) Predicted serine synthesis is increased in Tregs derived from cerebrospinal fluid of patients with treatment-naïve relapsing multiple sclerosis (MS) compared to controls (*Schafflick et al., 2020*). (**G**) Predicted serine synthesis is decreased in Tregs derived from prostate cancer versus control prostate (*Camps et al., 2023*). *p<0.05, **p<0.01, ****p<0.00001 by Student's t-test (**A, E, F, G**), Mann-Whitney U test (**C, D**), and two-way ANOVA with repeated measures (**B**).

## Synthesized and dietary serine regulate Treg function in vivo

After finding that serine derived from both extracellular sources and de novo synthesis inhibits Treg polarization in vitro, we investigated the effect on Tregs of altering serine availability in vivo. We first examined the impact of dietary serine restriction by maintaining mice immediately post-weaning on either a serine/glycine-free diet or isocaloric control diet. After 8 weeks, we measured Treg abundance in peripheral blood by flow cytometry. Mice fed a serine/glycine-free diet had significantly higher proportions of Tregs in peripheral blood (*Figure 3A*). To determine whether this effect on Tregs translates to protection in a murine model of autoimmunity, we used the myelin oligodendrocyte glycoprotein (MOG)$_{35-55}$ experimental autoimmune encephalomyelitis (EAE) model of multiple sclerosis (MS). Mice were transitioned to a serine/glycine-free or isocaloric control diet beginning one week prior to immunization with MOG$_{35-55}$, and disease severity was monitored by a blinded scorer.

A serine/glycine-free diet significantly attenuated disease severity, delaying the onset of neurologic deficits and reducing peak clinical scores (*Figure 3B–D*).

We next asked whether de novo serine synthesis is altered in human Tregs in disease. Using publicly available RNA-seq and scRNA-seq data, we estimated the rate of de novo serine synthesis using single-cell flux estimation analysis (SCFEA) (*Alghamdi et al., 2021*), which constructs a global set of metabolic reactions as a factor map and uses multi-layer neural networks to model reaction rates from mRNA levels of metabolic enzymes. We applied SCFEA to human Th0 and iTreg cells differentiated from naïve CD4 cells in vitro (*Ullah et al., 2018*) and found that serine synthesis was downregulated in iTregs (*Figure 3E*). Autoimmunity occurs when Tregs fail to adequately regulate the immune response, so we wondered whether the predicted rate of de novo serine synthesis would be altered in Tregs derived from patients with autoimmune disease. Using scRNA-seq data derived from Tregs isolated from the cerebrospinal fluid (CSF) of patients with treatment-naïve relapsing MS and controls (*Schafflick et al., 2020*), we found the predicted rate of serine synthesis was significantly higher in Tregs from MS patients (*Figure 3F*). Conversely, while autoimmunity results from inadequate Treg suppressive function, Tregs in the tumor microenvironment have pathologically high suppressive function. We examined scRNA-seq data from Tregs derived from healthy prostate and prostate cancer (*Tuong et al., 2021*) from the IMMUcan database (*Camps et al., 2023*) and found that prostate tumor-infiltrating Tregs had significantly lower predicted serine synthesis than Tregs from healthy prostate (*Figure 3G*). Consistent with our earlier findings that serine synthesis inhibits Treg function, these analyses suggest that serine synthesis within Tregs is higher in autoimmunity and lower in the tumor microenvironment, thereby negatively correlating with expected Treg suppressive function in human disease.

## Serine inhibits Treg generation by contributing to one-carbon metabolism and DNA methylation

We next examined the mechanistic links between serine metabolism and Treg function. Serine can serve as a one-carbon donor by entering the one-carbon metabolism cycle, which depends on conversion into glycine by the enzyme SHMT1/2 (*Reina-Campos et al., 2020*; *Figure 4A*). To determine whether the effects of serine on Treg differentiation depend on contributions to one-carbon metabolism, we cultured naïve murine CD4 cells under Treg polarizing conditions with either vehicle or the SHMT1/2 inhibitor SHIN1 (*Makino et al., 2022*; *Ducker et al., 2017*). Inhibiting SHMT1/2 increased Treg polarization (*Figure 4B*) and mitigated the inhibitory effect of extracellular serine on Tregs (*Figure 4C*). Because methionine can compensate as a one-carbon donor, other groups have found that the contribution of serine to one-carbon metabolism is greater in the absence of methionine (*Maddocks et al., 2016*). To further the idea that serine inhibits Treg polarization by contributing to one-carbon metabolism, we cultured naïve murine CD4 cells under Treg polarizing conditions in methionine-free media supplemented with either methionine or excess serine. Although methionine supplementation had little effect in the presence of physiologic extracellular serine (0.4 mM), excess extracellular serine (4 mM) was highly potent in inhibiting Treg polarization in the absence of methionine (*Figure 4D*). These studies suggest that the regulatory role of serine in Tregs is mediated at least in part through one-carbon metabolism.

A crucial role of one-carbon metabolism is to provide methyl groups for methylation reactions, and DNA methylation represents a key epigenetic mechanism used to silence gene transcription (*Mentch and Locasale, 2016*). We, therefore, asked whether serine entry into one-carbon metabolism contributes to DNA methylation in Tregs, so we conducted a carbon tracing experiment in which Tregs were cultured in media containing uniformly 13-carbon-labeled serine (U$^{13}$C-serine) for 12 hr before DNA was isolated from the unsorted cells and analyzed by mass spectrometry. We found that serine directly contributed to cytosine methylation, as measured by carbon labeling of methylcytosine (*Figure 4E*). Demethylation of the Treg-specific demethylated region (TSDR) in Tregs is a critical step for the generation of stable, bona fide Tregs, and methylation in this region is associated with transient or unstable expression of FOXP3 and suppressive activity (*Polansky et al., 2008*). We, therefore, investigated the effects of extracellular serine and PGAM knockdown on methylation of the TSDR via pyrosequencing. In a first set of experiments, naïve murine CD4 cells were cultured under Treg polarizing conditions in either serine/glycine-free media or media containing serine. The presence of extracellular serine increased TSDR methylation (*Figure 4F*). Genetic knockdown of PGAM using ASOs similarly increased

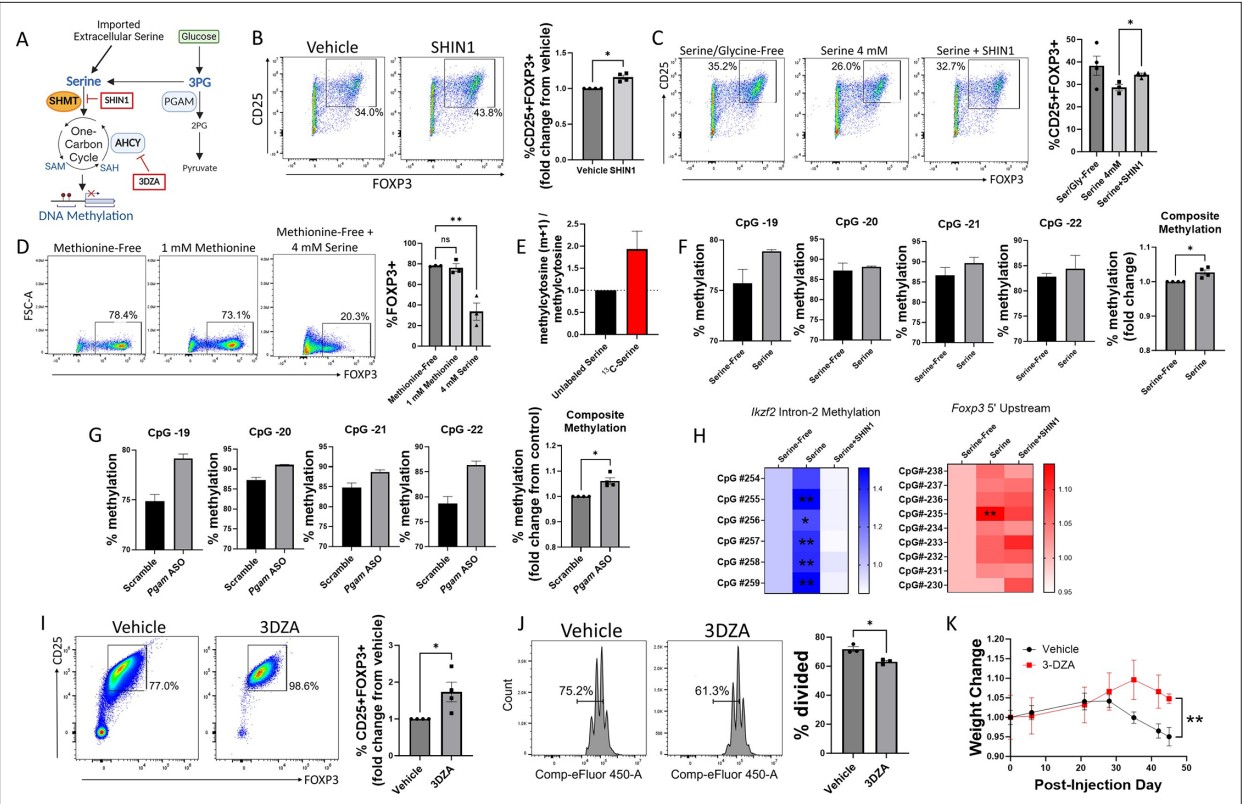

**Figure 4.** Serine inhibits Treg polarization by contributing to one-carbon metabolism and methylation of Treg-associated genes. (**A**) Schematic diagram of serine entry into one-carbon metabolism and the methyl donor cycle. Pharmacologic inhibitors are shown in red boxes. This panel was created using BioRender.com. (**B**) Naïve murine CD4 cells were cultured under Treg polarizing conditions for 72 hr with either vehicle or SHIN1 (SHMT1/2 inhibitor, 1 µM) added at day 0 of culture. Treg polarization was assayed by flow cytometry. Data represent mean ± SEM from four independent experiments, with 3–4 biological replicates per experiment. (**C**) Naïve murine CD4 cells were cultured in serine/glycine-free media, media containing 4 mM serine, or media with serine plus SHIN1 (1 µM). Treg polarization was assayed by flow cytometry. Data represent mean ± SEM from three to four biological replicates per condition. (**D**) Naïve murine CD4 cells were cultured in methionine-free media supplemented with vehicle, 1 mM methionine, or 4 mM serine, and FOXP3 expression was assayed by flow cytometry. Data represent mean ± SEM from three biological replicates. (**E**) Naïve murine CD4 cells were cultured in serine/glycine-free media for 48 hr under Treg polarizing conditions and then for 12 hr with either unlabeled serine or U$^{13}$C-serine. The mass isotopomer ratio of carbon-labeled methylcytosine (m+1) / unlabeled methylcytosine from unsorted cells is shown. Data represent mean ± SEM from three biological replicates. (**F**) Naïve CD4 cells were cultured under Treg polarizing conditions with either serine/glycine-free media or 4 mM serine. Methylation at the *Foxp3* TSDR was assayed by pyrosequencing from unsorted cells. The composite methylation was calculated as the percent methylation for each of the 4 CpG sites measured in the assay normalized to the serine/glycine-free condition. Data derived from three biological replicates per condition. (**G**) Naïve CD4 cells were cultured under Treg polarizing conditions with either scrambled or anti-*Pgam* ASOs. Methylation at the *Foxp3* TSDR was assayed by pyrosequencing from unsorted cells. The composite methylation was calculated as the percent methylation at each of the 4 CpG sites measured in the assay normalized to scrambled control. Data derived from three biological replicates per condition. (**H**) Naïve murine CD4 cells were cultured in serine/glycine-free media, media containing 4 mM serine, or media with serine plus SHIN1 (1 µM). Bisulfite-treated DNA was sequenced from unsorted cells using a Treg-specific next generation sequencing panel, and percent methylation of key sites was calculated. Data derived from three biological replicates per condition. (**I**) Naïve CD4 cells were cultured under Treg polarizing conditions with either vehicle or 3-Deazaadenosine (3DZA, AHCY inhibitor, 5 µM). Treg polarization was assayed by flow cytometry. Data represent mean ± SEM from 4 independent experiments, with three to four biological replicates per experiment. (**J**) Naïve CD4 cells were polarized to Tregs with either vehicle or 3DZA and then cultured with CD4 cells stimulated with CD3/CD28 to evaluate Treg suppressive function. CD4 cell proliferation was measured by flow cytometry. Data represent mean ± SEM from 3 biological replicates. (**K**) Naïve CD4 cells were cultured under Treg polarizing conditions with either vehicle or 3DZA, and 10$^6$ cells were then transferred into RAG -/- mouse recipients along with 10$^7$ activated CD4 cells. Mice were weighed at the specified times and weights are shown relative to the weight at the day of injection. Data represent mean ± SEM from five mice per group. *p<0.05, **p<0.01 by Mann-Whitney U test (**B, F, G, I**), one-way ANOVA with Tukey's multiple comparisons testing (**C, D**), two-way ANOVA with Dunnett's multiple comparison testing (**H**), Student's t-test of average final weight (**K**), or Student's t-test (**J**).

The online version of this article includes the following figure supplement(s) for figure 4:

**Figure supplement 1.** 3DZA does not affect viability.

**Figure supplement 2.** Tregs polarized with 3DZA are more suppressive in the T cell transfer model of autoimmune colitis.

TSDR methylation (*Figure 4G*). Taken together, these findings demonstrate that both extracellular serine and PGAM-regulated serine synthesis increase the methylation, and therefore silencing, of Treg-associated genes.

Given the promising results seen with pyrosequencing, we next used a next generation sequencing panel to identify CpG sites in Treg-specific genes whose methylation is regulated by serine (*Figure 4H*). Methylation of *Ikzf2* (Helios) was significantly increased with the addition of extracellular serine, but this was reversed by blocking serine entry into one-carbon metabolism with the SHMT inhibitor SHIN1. In the 5' upstream region of the *Foxp3* gene, one CpG site showed significantly higher methylation in the presence of serine, and this effect was reversed by SHIN1. A complete list of CpG methylation levels can be found in *Supplementary file 2*.

To definitively link methylation to Treg differentiation and suppressive function, we blocked methylation in Tregs with 3-Deazaadenosine (3DZA), an inhibitor of S-adenosylhomocysteine (SAH) hydrolase. SAH hydrolase degrades SAH and prevents recycling of the methyl donor S-adenosylmethionine (SAM) (*Jeong et al., 1999*). In naïve murine CD4 cells cultured under Treg polarizing conditions, 3DZA increased Treg differentiation (*Figure 4I*) without affecting cell viability (*Figure 4—figure supplement 1*). Treatment with 3DZA similarly augmented suppressive function in cultured murine Tregs (*Figure 4J*). To determine the in vivo relevance of these findings, we used an adoptive transfer RAG colitis mouse model. Conventional murine CD4 cells were activated ex vivo and injected into RAG$^{-/-}$ mice to induce colitis, along with simultaneous injection of Tregs that had been treated during differentiation with either vehicle or 3DZA. Mice injected with Tregs polarized in the presence of 3DZA had less severe disease as measured by weight loss and colonic damage (*Figure 4K*, *Figure 4—figure supplement 2*), indicating greater Treg suppressive function. Altogether, these findings demonstrate that one-carbon metabolism and methylation impair Treg differentiation and suppressive function both in vitro and in vivo.

## Discussion

In this study, we found that PGAM plays a key physiologic role in Treg differentiation and suppressive function through regulation of serine synthesis, which intersects with glycolysis immediately upstream of PGAM through the shared substrate 3 PG. Analysis of published datasets revealed that PGAM expression is specifically increased within human Tregs at both transcriptional and protein levels, and further that PGAM expression is enriched in highly suppressive Treg phenotypes. Pharmacologic inhibition or genetic knockdown of PGAM reduced Treg polarization and suppressive function while promoting a pro-inflammatory gene signature. These effects were replicated by exogenous supplementation of 3 PG and reversed by inhibition of serine synthesis. Single-cell flux estimation analysis predicted increased serine synthesis in Tregs isolated from patients suffering from autoimmune disease and decreased serine flux in tumor-associated Tregs, underlining the clinical importance of this regulatory pathway. Consistent with a previous report (*Kurniawan et al., 2020*), we found that both synthesized and exogenously imported serine impact Treg function. Mechanistically, serine inhibited Treg polarization by increasing flux through one-carbon metabolism with downstream effects on DNA methylation of Treg-related genes. Interfering with serine availability or methylation attenuated disease in in vivo models of autoimmunity.

The surprising role of PGAM in Treg differentiation has both physiologic and therapeutic implications. Although glycolysis has been shown to support initial proliferation and migration of Tregs (*Kishore et al., 2017*; *De Rosa et al., 2015*), most prior work has suggested that glycolytic flux supports effector T cell functions with concomitant downregulation of Treg stability and suppressive activity (*Gerriets et al., 2015*; *Michalek et al., 2011*; *Macintyre et al., 2014*; *Shi et al., 2011*). Indeed, previous studies have shown Tregs to depend on fatty acid oxidation and oxidative phosphorylation, largely independent of glycolysis (*Angelin et al., 2017*; *Lee et al., 2014*; *Berod et al., 2014*). Our own prior work found that inhibition of GAPDH, an enzyme two catalytic reactions upstream of PGAM in glycolysis, enhances Treg differentiation (*Kornberg et al., 2018*), with similar findings observed following inhibition or knockdown of other proximal glycolytic enzymes (*Shi et al., 2011*; *Scherlinger et al., 2022*; *Tanimine et al., 2019*). It is striking, therefore, that PGAM enzymatic activity has the opposite effect. Inhibition or knockdown of PGAM not only attenuated Treg differentiation and suppressive function but reciprocally enhanced markers of a pro-inflammatory, Th17-like phenotype, implicating PGAM as a physiologic regulator of the balance between pro- and anti-inflammatory

lymphocyte responses via regulation of serine synthesis. These findings suggest that glycolytic regulation of immunity is more complex than previously appreciated, supporting a more granular view in which glycolytic enzymes can serve as independent regulatory nodes for the fine-tuning of immune responses. Our data suggest that these differential effects of glycolytic enzymes might be mediated by fine control of the metabolites whose concentrations they regulate, such as 3 PG in the case of PGAM.

From a therapeutic perspective, our findings identify novel targets for modulating Treg function in disease. Substantial prior work has shown that manipulation of glycolysis and other metabolic pathways has a therapeutic window for systemic administration despite the ubiquity of these pathways in cell biology (*Patel et al., 2019*). Nonetheless, systemic administration may not be necessary for several key therapeutic applications, for instance, optimization of chimeric antigen receptor (CAR)-T cell therapies (*McPhedran et al., 2024*; *Nanjireddy et al., 2023*; *Chan et al., 2024*). Differentiation of CAR-T cells into a Treg-like state correlates with failure of CAR-T therapy in B cell lymphoma (*Good et al., 2022*), such that manipulation of PGAM and downstream serine metabolism and DNA methylation is a potential strategy for improving the anti-cancer fitness of CAR-T cells ex vivo prior to infusion. Conversely, CAR-Tregs are currently in clinical trials for a variety of autoimmune disorders (*Zhang et al., 2018*; *Abraham et al., 2023*), and there is widespread interest in treatments that improve the suppressive activity of Tregs.

Of note, a prior study reported that complete knockout of PGAM1 in T cells impaired differentiation and effector functions of pro-inflammatory T cell subsets with only a mild impact on Treg generation (*Toriyama et al., 2020*). However, complete knockout of a glycolytic enzyme would be expected to abrogate glycolytic flux entirely, with wide-ranging effects (including decreased energy production) likely to mask the impact of physiologic or therapeutic modulation of enzyme function. As we showed here, partial knockdown or pharmacologic inhibition of PGAM1, which decreases but does not fully abrogate enzymatic activity, substantially reduced Treg differentiation while supporting pro-inflammatory activation. Additionally, other groups studying glycolysis in cancer used partial knockdowns to show that PGAM supports tumor proliferation by controlling flux through the de novo serine synthesis pathway (*Itoyama et al., 2021*).

Our investigation focused on the contribution of one-carbon metabolism to DNA methylation, which is particularly relevant in Tregs given the established regulatory role of methylation of Treg-specific genes, including the Treg Specific Demethylated Region (TSDR) of the FOXP3 gene locus (*Zafari et al., 2018*; *Toker et al., 2013*; *Kressler et al., 2020*). However, one-carbon metabolism contributes to many cellular processes, including purine synthesis, amino acid metabolism, and phospholipid synthesis, among others (*Mentch and Locasale, 2016*), so it is likely that one-carbon metabolism inhibits Treg differentiation in a variety of ways. Although other groups have also identified the importance of one-carbon metabolism in Treg generation (*Sugiura et al., 2022*), we describe novel links between glycolysis, serine synthesis, and DNA methylation as a regulatory pathway of Treg differentiation.

In summary, we have found an unexpected role of PGAM that adds further nuance to the complex relationship between glycolysis and Treg function and further implicates serine and its contributions to one-carbon metabolism and DNA methylation as key regulators of Treg differentiation. In addition to yielding new biological insights into the metabolic regulation of lymphocyte fate and function, these findings point to new strategies for the therapeutic manipulation of Treg function.

# Materials and methods

## Key resources table

| Reagent type (species) or resource | Designation | Source or reference | Identifiers | Additional information |
|---|---|---|---|---|
| Gene (*Homo sapiens*) | PGAM1 | GenBank | Gene ID: 5223 | |
| Gene (*Mus musculus*) | Pgam1 | GenBank | Gene ID: 18648 | |
| Strain, strain background (*Mus musculus*, male and female) | C57BL/6 J wild-type | Jackson Laboratory | Cat # 000664, RRID:IMSR_JAX:000664 | |

*Continued on next page*

*Continued*

| Reagent type (species) or resource | Designation | Source or reference | Identifiers | Additional information |
|---|---|---|---|---|
| Strain, strain background (*Mus musculus*, male, and female) | B6.129S7-*Rag1tm1Mom*/J (RAG knockout mice) | Jackson Laboratory | Cat # 002216, RRID:IMSR_JAX:002216 | |
| Antibody | Anti-mouse CD25, BV785 (Rat monoclonal) | Biolegend | Cat # 102051, RRID:AB_2564131 | 1:200 |
| Antibody | Anti-FOXP3, APC (Rat monoclonal) | Thermo Fisher | Cat # 17-5773-80, RRID:AB_469456 | 1:200 |
| Antibody | Anti-PGAM1, Alexa Fluor 405 (Rabbit polyclonal) | Novus Biologicals | Cat # NBP1-49532AF405, RRID:AB_3212871 | 1:200 |
| Antibody | Anti-CD3 (OKT3), Functional Grade, eBioscience (Mouse monoclonal) | Thermo Fisher | Cat # 16-0037-81, RRID:AB_468854 | 2 µg/ml |
| Antibody | Anti-CD28 (37.51), Functional Grade, eBioscience (Syrian hamster monoclonal) | Thermo Fisher | Cat # 16-0281-82, RRID:AB_468921 | 2 µg/ml |
| Antibody | *InVivo*Plus anti-mouse IFNγ (Rat monoclonal) | BioXcell | Cat # BP0055, RRID:AB_1107694 | 5 µg/ml |
| Antibody | *InVivo*Plus anti-mouse IL4 (Rat monoclonal) | BioXcell | Cat # BE0045, RRID:AB_1107707 | 5 µg/ml |
| Commercial assay or kit | Zombie NIR fixable viability kit | BioLegend | Cat # 423106 | 1:1000 |
| Commercial assay or kit | Cell Proliferation Dye | eBioscience | Cat # 65-0842-85 | 1:2000 |
| Commercial assay or kit | MojoSort(TM) Mouse CD4 Naive T Cell Isolation Kit | BioLegend | Cat # 480040 | |
| Commercial assay or kit | MojoSort(TM) Mouse CD4 T Cell Isolation Kit | BioLegend | Cat # 480006 | |
| Chemical compound, drug | SHIN1 | MCE | Cat # HY-112066 | |
| Chemical compound, drug | NCT-503 | Sigma | Cat # SML1659-5MG | |
| Chemical compound, drug | EGCG | Millipore Sigma | Cat # E4143-50MG | |
| Chemical compound, drug | 3-DZA | Cayman | Cat # 9000785 | |
| Chemical compound, drug | Heptelidic Acid | Cayman | Cat # 14079 | |
| Chemical compound, drug | D-(-)–3-Phosphoglyceric Acid (Sodium Salt) | Cayman | Cat # 20123 | |
| Chemical compound, drug | D-(+)-Glucose-13C6 | Cayman | Cat # 26707 | |
| Chemical compound, drug | L-Serine-13C3 | Cayman | Cat # 35126 | |
| Sequence-based reagent | *Pgam* antisense oligonucleotides | Aum Biotech | Custom order | |
| Sequence-based reagent | *Phgdh* antisense oligonucleotides | Aum Biotech | Custom order | |
| Peptide, recombinant protein | Animal-Free Recombinant Murine IL-2 | Thermo Fisher | Cat # AF-212–12 | |
| Peptide, recombinant protein | Recombinant Human TGF-beta 1 | BioRad | Cat # PHP143B | |
| Peptide, recombinant protein | Murine IL-6 | Peprotech | Cat # 216–16 | |
| Peptide, recombinant protein | Recombinant Mouse IL-23 (carrier-free) | BioLegend | Cat # 589002 | |
| Peptide, recombinant protein | Murine IL-1beta | Peprotech | Cat # 211-11B | |
| Other | Control mouse diet | Animal Specialties and Provisions | Cat # 1817070–209 | |
| Other | Serine/Glycine-Free mouse diet | Animal Specialties and Provisions | Cat # 1812281 | |

## Resource availability
### Lead contact
Further information and requests for resources and reagents should be directed to and will be fulfilled by the Lead Contact, Michael D. Kornberg (michael.kornberg@jhmi.edu).

## Materials availability
This study did not generate new unique reagents.

## Experimental model and subject details

### Animals

Wild-type C57BL/6 J (stock # 000664) and B6.129S7-*Rag1*<sup>tm1Mom</sup>/J RAG knockout mice (stock # 002216) were purchased from The Jackson Laboratory. All mice were housed in a dedicated Johns Hopkins mouse facility. All protocols were approved by the Johns Hopkins Institutional Animal Care and Use Committee, protocol numbers MO21M370, and MO24M348.

## Primary cell cultures

### CD4+ T-cell isolation and culture

T-cells were isolated by generating single-cell suspensions from the spleen and lymph nodes of 6–10 week-old C57BL/6 mice. Spleens were mechanically dissociated with the plunger of a syringe over a 100 µm cell strainer (BD Falcon), then were pelleted by centrifugation (1500 RPM for 5 min) and resuspended in fresh FACS buffer. Naive CD4 + cells were isolated by negative selection using the MojoSort CD4 + naïve T-cell isolation kit from Biolegend (Catalog 480040) according to the manufacturer's instructions. Plates were coated overnight at 4 °C with plate-bound anti-mouse CD3 antibody (2 µg/mL). Prior to plating, cells were resuspended in complete 50% RPMI (RPMI plus 10% FBS, 1% Pen-strep, 1% Glutamax, 0.1% 2-mercaptoethanol) and 50% AIM-V media along with 2 µg/mL soluble anti-mouse CD28 antibody. Experiments involving serine/glycine-free media or methionine-free media were done with 100% complete RPMI lacking the corresponding amino acid(s). Treg polarizing conditions included 10 ng/mL murine IL-2 and 1 ng/mL human TGF-β along with 5 µg/mL anti-mouse IFNγ and 5 µg/mL anti-mouse IL-4. Experiments with Treg polarization for Treg suppression assays were performed with 15 ng/mL human TGF-β and 10 ng/mL retinoic acid. Cells were not sorted prior to downstream assays, although bulk experiments were only performed on samples with greater than 70% cell viability (greater than 90% for stable isotope tracing studies).

## Method details

### Drug and ASO treatments

For drug treatments, cells were treated with the indicated drug at the indicated concentration. EGCG was dissolved in EtOH, while NCT-503 and SHIN1 were dissolved in DMSO. Vehicle controls for drug treatment experiments consisted of solvent alone. Antisense oligonucleotides (ASOs) were dissolved in PBS and added at 10 µM directly to the cell culture media starting at day 0. Control samples for ASO experiments were treated identically, but with scrambled ASOs.

### Electroporation

For electroporation with 3-phosphoglycerate (3 PG), naïve CD4 cells were cultured for 24 hr under Treg polarizing conditions, and pre-treated with NCT-503 or vehicle for 4 hr prior to electroporation as indicated. Cells were then harvested, centrifuged, and resuspended in BTX high performance electroporation solution (45–0803) with either 1.5 mM 3 PG or PBS as indicated. Electroporation was performed at 500 V for 3 ms with 1 pulse/square in a 4 mM BTX electroporation cuvette. Cells were then immediately added to complete RPMI media that contained 1.5 mM 3 PG or PBS and rested for 2–4 hr. Cells were then centrifuged and resuspended in the media they were in prior to electroporation.

### Flow cytometry

Cells were stained with Zombie NIR (1:1000) for 10 min and then washed with Flow Wash Buffer (PBS + 2% fetal bovine serum +2 mM EDTA). They were then stained with anti-CD25 (BV785) for 30 min. Cells were then washed with Flow Wash buffer two times and fixed with FoxP3 fixation/permeabilization buffer following the manufacturer's recommended protocol. Intracellular staining was performed with conjugated antibodies (1:200) against the specified proteins in permeabilization buffer for 1 hr, washed twice, and then analyzed cells with a Cytek Aurora Flow cytometer. Data analysis was performed with FlowJo software.

## Treg suppression assay

To generate Tregs for the Treg suppression assay, naïve CD4 T cells were cultured for 4 days under Treg polarizing conditions. Then, total CD4 + T cells were isolated from murine spleens using Biolegend Mojo CD4 T cell isolation kit (480006) and labelled with Cell Proliferation Dye (eBioscience, #65-0842-85) by incubating in a 1:2000 dilution (PBS) for 10 min, washing 1 X with PBS, and then washing 2 X with complete RPMI. CD4 + T cells were cultured as described above (3 ug/mL anti-CD3, 3 ug/mL anti-CD28, 10 ng/mL recombinant murine IL-2) with varying ratios of Treg:T effector cells. Divisions were counted by measuring cell proliferation dye dilutions with flow cytometry after a 3 day culture.

## Mouse peripheral blood samples

Mice were fed either a serine/glycine-free diet or a control diet (Animal Specialty and Provisions, 1812281 and 1817070–209) for 8 weeks immediately post-weaning and peripheral blood samples were obtained via cheek bleed into a microfuge tube coated with heparin/EDTA. RBCs were lysed with RBC lysis buffer for 7 min and then the reaction was stopped with HBSS (including Calcium and Magnesium). Cells were then stained with Zombie NIR (1:1000) and anti-CD16/32 (1:200) for 10 min, then stained with anti-CD45, anti-CD11b, and anti-CD4 for 30 min. Staining then proceeded as described in the flow cytometry section.

## Mass spectrometry

Sample preparation for steady-state measurement of serine involved culturing CD4 T cells for 72 hr under Treg polarizing conditions, followed by treatment with the indicated drugs for 24 hr. The cells were collected, washed with PBS, and equal number of cells per condition were pelleted and flash-frozen. The frozen cell pellets were shipped to the Metabolomics Core at Baylor College of Medicine, Houston, TX. The cells were then resuspended in a 500 µl mixture of 1:1 water/methanol, sonicated for 1 min (two 30 s pulses), and mixed with 450 µl of ice-cold chloroform. The resulting homogenate was combined with 150 µl of ice-cold water, vortexed for 5 min, incubated at –20 °C for 30 min, and centrifuged at 4 °C for 10 min to partition the aqueous and organic layers. The combined layers were dried at 37 °C for 45 min in a SpeedVac system (Genevac EZ Solvent Evaporator, SP Industries, Inc). The extract was reconstituted in a 500 µl solution of methanol/water (1:1) and filtered through a 3 kDa molecular filter (Amicon Ultra Centrifugal Filter, 3 kDa MWCO, Millipore Sigma) at 4 °C for 90 min to remove proteins. The filtrate was dried at 37 °C for 90 min in a SpeedVac and stored at –80 °C until mass spectrometric analysis. Prior to mass spectrometric analysis, the dried extract was resuspended in methanol/water (1:1).

For amino acid flux analysis, CD4 T cells were cultured for 6 hr in glucose-free media, then 12 mM U13C-glucose was added for 6 hr. The cells were resuspended in a 500 µl mixture of 1:1 water/methanol, sonicated for 1 min (two 30 s pulses), centrifuged at 4 °C for 10 min, and filtered through a 3 kDa molecular filter (Amicon Ultra Centrifugal Filter, 3 kDa MWCO, Millipore Sigma) at 4 °C for 90 min to remove proteins. The samples were dried at 37 °C for 45 min in a SpeedVac system (Genevac EZ Solvent Evaporator, SP Industries, Inc). Prior to mass spectrometry analysis, the dried extract was resuspended in methanol/water (1:1).

CD4 T cells were cultured under Treg polarizing conditions in serine-free media to measure methyl-cytosine flux, followed by the addition of either normal serine or 13C3-serine for 12 hr. The cells were harvested, the pellets were washed, and DNA was isolated using the Qiagen DNeasy Blood & Tissue Kit. DNA was quantified with a Nanodrop, and all samples were normalized based on total DNA content before mass spectrometry analysis. The details of the DNA hydrolysis methods were previously described (PMID: 27158554). DNA was denatured at 100 °C for 3 min, incubated with ammonium acetate and nuclease P1, then mixed and incubated with venom phosphodiesterase I, followed by incubation with alkaline phosphatase. The details of the liquid chromatography-mass spectrometry (LC-MS) methods were also described earlier (PMID: 27158554).

For the LC-MS analysis method for serine steady state and tracing study of serine and methyl-cytosine, chromatographic separation of extracted metabolites, including serine, was performed through Hydrophilic Interaction Chromatography (HILIC) using an XBridge Amide column (3.5 µm, 4.6×100 mm, Waters, Milford, MA) in ESI-positive ionization mode. Solvent A (0.1% formic acid in water) and Solvent B (0.1% formic acid in acetonitrile) were used for chromatographic separation (Ref). The data were acquired via multiple reaction monitoring (MRM) using a 6495 Triple Quadrupole mass

spectrometer coupled to an HPLC system (Agilent Technologies, Santa Clara, CA) through Agilent Mass Hunter Data Acquisition Software (ver. 10.1) (PMID: 31360899, PMID: 30642841). The acquired mass spectra were analyzed and integrated into each targeted compound peak using Agilent Mass Hunter Quantitative Analysis Software. For serine measurement, peak areas were normalized with a spiked internal standard. Differential metabolites were determined using p-values from t-tests following the Benjamini-Hochberg method, with a false discovery rate (FDR) of less than 0.25. For serine and methylcytosine tracing studies, the percentage of 13 C incorporation was calculated using Microsoft Excel from peak areas and represented as bar graphs.

## EAE induction and scoring

Active EAE was induced in female and male C57BL/6J mice (*Abramson et al., 2021*; *Gharibani et al., 2025*; *Godfrey et al., 2022*) (10 weeks old) after pretreatment with either serine/glycine-free diet or control diet (Animal Specialty and Provisions, 1812281 and 1817070–209) for 1 week. $MOG_{35-55}$ peptide dissolved in PBS at a concentration of 2 mg/mL was mixed 1:1 with complete Freund's adjuvant (8 mg/ml Tuberculin toxin in incomplete Freund's adjuvant), then mixed for 12 min into an emulsion. On day 0, mice were immunized by injecting 75 µl of the emulsion subcutaneously into each of two sites on the lateral abdomen. In addition, on day 0 and again on day 2, mice were injected intraperitoneally with 375 ng pertussis toxin. Scoring was performed by a blinded individual according to the following scale: 0, no clinical deficit; 0.5, partial loss of tail tone; 1.0, complete tail paralysis or both partial loss of tail tone plus awkward gait; 1.5, complete tail paralysis and awkward gait; 2.0, tail paralysis with hind limb weakness evidenced by foot dropping between bars of cage lid while walking; 2.5, hind limb paralysis with little to no weight-bearing on hind limbs (dragging), but with some movement possible in legs; 3.0, complete hind limb paralysis with no movement in lower limbs; 3.5, hind limb paralysis with some weakness in forelimbs; 4.0, complete tetraplegia but with some movement of head; 4.5, moribund; and 5.0, dead.

## RNA-seq

For all experiments, RNA was isolated using the Qiagen RNeasy Mini Plus Kit. For RNA-seq, library preparation, sequencing, and demultiplexing were performed by Novogene. Raw FASTQ files were aligned to the murine reference genome with Kallisto (*Bray et al., 2016*). Library size normalization and differential expression was performed with DESeq2 (*Love et al., 2014*; *Muzellec et al., 2023*). Pathway enrichment analysis was performed with GSEA (*Reimand et al., 2019*; *Fang et al., 2023*), and hypergeometric tests were performed with EnrichR (*Kuleshov et al., 2016*).

## Single cell flux estimation analysis (SCFEA)

Re-analysis of publicly available data was performed by downloading the relevant dataset from the GEO repository or from figshare. Gene count matrices were extracted and then analyzed with the SCFEA python package or web server. The rate of de novo serine synthesis was estimated with the reaction titled '3PG -> Serine.' Comparisons between groups were performed with either Student's t-test or the Wilcoxon Rank Sum test as described in the figure legends.

## Bisulfite conversion and pyrosequencing

Bisulfite conversion was performed by EpigenDx according to standard pyrosequencing and Next Gen Sequencing protocols. Briefly, cell pellets were lysed based on the total cell count per sample and total volume received using M-digestion Buffer (ZymoResearch; Irvine, CA; cat# D5021-9) and 5–10 µL of protease K (20 mg/mL), with a final M-digestion concentration of 1 X. The samples were incubated at 65 °C for a minimum of 2 hr. For bisulfite modification, 20 µL of the supernatant from the sample extracts was bisulfite modified using the EZ-96 DNA Methylation-Direct Kit (ZymoResearch; Irvine, CA; cat# D5023) as per the manufacturer's protocol. The bisulfite-modified DNA samples were eluted using Melution buffer in 46 µL. For Multiplex PCR, all bisulfite-modified DNA samples were amplified using separate multiplex or simplex PCRs. PCRs included 0.5 units of HotStarTaq (Qiagen; Hilden, Germany; cat# 203205), 0.2 µM primers, and 3 µL of bisulfite-treated DNA in a 20 µL reaction. All PCR products were verified using the Qiagen QIAxcel Advanced System (v1.0.6). Prior to library preparation, PCR products from the same sample were pooled and then purified using the

QIAquick PCR Purification Kit columns or plates (cat# 28106 or 28183). The PCR cycling conditions were: 95 °C 15 min; 45 x (95 °C 30 s; Ta°C 30 s; 68 °C 30 s); 68 °C 5 min; 4 °C ∞. Samples were run alongside established reference DNA samples with a range of methylation. They were created by mixing high- and low-methylated DNA to obtain samples with 0, 50, and 100% methylation. The high-methylated DNA was in vitro enzymatically methylated genomic DNA with >85% methylation. The low-methylated DNA was chemically and enzymatically treated with <5% methylation. They were first tested on numerous gene-specific and global methylation assays using pyrosequencing.

## Methylation sequencing

Libraries were prepared using a custom Library Preparation method created by EpigenDx. Next, library molecules were purified using Agencourt AMPure XP beads (Beckman Coulter; Brea, CA; cat# A63882). Barcoded samples were then pooled in an equimolar fashion before template preparation and enrichment were performed on the Ion Chef system using Ion 520 & Ion 530 ExT Chef reagents (Thermo Fisher; Waltham, MA; cat# A30670). Following this, enriched, template-positive library molecules were sequenced on the Ion S5 sequencer using an Ion 530 sequencing chip (cat# A27764). FASTQ files from the Ion Torrent S5 server were aligned to a local reference database using the open-source Bismark Bisulfite Read Mapper program (v0.12.2) with the Bowtie2 alignment algorithm (v2.2.3). Methylation levels were calculated in Bismark by dividing the number of methylated reads by the total number of reads. An R-squared value (RSQ) was previously calculated from controls set at known methylation levels to test for PCR bias.

## T-cell transfer model of inflammatory bowel disease

To induce autoimmune colitis, CD4 T cells were transferred into RAG-knockout mice (Jackson Laboratories #002216). Briefly, CD4 cells were isolated from wild-type C57BL/6 mice and cultured in a mixture of 50% cRPMI (RPMI +10% Fetal Bovine Serum, 1% penicillin-streptomycin, and 0.1% 2-Mercaptoethanol) and 50% AIM-V media with plate bound anti-CD3 stimulation (3 µg/mL) and anti-CD28 stimulation (2 µg/mL) along with 10 ng/mL murine IL-2. Cells were stimulated for 3 days, rested for 2 days, and then re-stimulated for 2 days. 10 million activated cells were then transferred into each mouse via i.p. injection. At the same time, naïve CD4 cells were cultured under Treg polarizing conditions as described above and 1 million cells were transferred into each mouse at the same time as the injection of the activated cells. Weights were recorded weekly until euthanasia.

## Quantification and statistical analysis

All statistical analyses were performed using GraphPad Prism software, Python, or R Studio. Details of statistical analyses for each experiment and/or data set can be found in the figures and figure legends. When cumulative data was tabulated from multiple independent experiments, experimental groups from each independent experiment were normalized to a matched control condition to allow integration across experiments. In these cases, in which there is no variability among matched control values, statistical analysis was performed with a non-parametric test as outlined in standard statistical guidelines (doi: 10.1111/bph.14153).

## Acknowledgements

We kindly thank Matthew Smith, Payam Gharibani, Erika Pearce, Ed Pearce, Brett Morrison, and Peter Calabresi for constructive comments and technical assistance. Additionally, we thank EpigenDx for their help with methylation NGS panels and pyrosequencing. *Figures 2A and 4A* were made using Biorender.com. This work was supported by NIH/NINDS Grant K08NS104266, Conrad N Hilton Foundation Marilyn Hilton Bridging Award for Physician Scientists Grant 17316, and a Herbert R and Jeanne C Mayer Foundation grant to MDK. WHG was supported by NIH MSTP Grant T32 GM136577 and a Careers in Immunology Fellowship Award from the American Association of Immunologists. Metabolomics and mass spectrometry work was supported by the following awards to NP: CPRIT Proteomics and Metabolomics Core Facility (RP210227), NIH (P30 CA125123), NIH (R01CA282282), and Dan L Duncan Cancer Center.

# Additional information

## Competing interests

Michael D Kornberg: Has received ad hoc consulting fees from Biogen Idec, Genentech, Janssen Pharmaceuticals, Novartis, OptumRx, and TG Therapeutics on topics unrelated to this manuscript. The other authors declare that no competing interests exist.

## Funding

| Funder | Grant reference number | Author |
| --- | --- | --- |
| National Institutes of Health | K08NS104266 | Michael D Kornberg |
| Conrad N. Hilton Foundation | Marilyn Hilton Bridging Award for Physician Scientists 17316 | Michael D Kornberg |
| Herbert R. and Jeanne C. Mayer Foundation | | Michael D Kornberg |
| National Institutes of Health | T32 GM136577 | Wesley H Godfrey |
| American Association of Immunologists | Careers in Immunology Fellowship Award | Wesley H Godfrey |
| National Institutes of Health | P30 CA125123 | Nagireddy Putluri |
| National Institutes of Health | R01CA282282 | Nagireddy Putluri |
| CPRIT Proteomics and Metabolomics Core Facility | RP210227 | Nagireddy Putluri |
| Dan L. Duncan Cancer Center | | Nagireddy Putluri |

The funders had no role in study design, data collection and interpretation, or the decision to submit the work for publication.

## Author contributions

Wesley H Godfrey, Conceptualization, Data curation, Formal analysis, Investigation, Visualization, Methodology, Writing – original draft, Writing – review and editing; Judy J Lee, Formal analysis, Investigation; Shruthi Shanmukha, Kaho Cho, Xiaojing Deng, Chandra Shekar R Ambati, Investigation; Vasanta Putluri, Abu Hena Mostafa Kamal, Formal analysis, Investigation, Visualization; Paul M Kim, Supervision, Writing – review and editing; Nagireddy Putluri, Formal analysis, Funding acquisition, Visualization, Methodology, Writing – review and editing; Michael D Kornberg, Conceptualization, Data curation, Formal analysis, Supervision, Funding acquisition, Writing – original draft, Writing – review and editing

## Author ORCIDs

Wesley H Godfrey ⓘ https://orcid.org/0000-0002-0834-492X
Judy J Lee ⓘ https://orcid.org/0000-0001-9862-3502
Shruthi Shanmukha ⓘ https://orcid.org/0000-0001-5659-2250
Paul M Kim ⓘ https://orcid.org/0000-0002-9150-9536
Nagireddy Putluri ⓘ https://orcid.org/0000-0003-4488-7400
Michael D Kornberg ⓘ https://orcid.org/0000-0003-2588-7325

## Ethics

All mice were housed in a pathogen-free, temperature-controlled Johns Hopkins (JH) mouse facility. All protocols were approved by the JH Institutional Animal Care and Use Committee (protocol numbers MO21M370 and MO24M348).

Reviewer #1 (Public review): https://doi.org/10.7554/eLife.104423.3.sa1

Reviewer #2 (Public review): https://doi.org/10.7554/eLife.104423.3.sa2
Author response https://doi.org/10.7554/eLife.104423.3.sa3

## Additional files

### Supplementary files
Supplementary file 1. List of differentially expressed genes -*Pgam* ASOs versus Scrambled ASOs.
Supplementary file 2. CpG methylation of Treg-specific genes by next generation sequencing.
MDAR checklist

### Data availability
All data supporting the findings of this study are presented and/or tabulated within the article or supplementary files. RNA-sequencing data from Figures 1F-1H has been deposited in the Gene Expression Omnibus (GEO) database and can be accessed via GEO accession number GSE269846. Other source data has been deposited in Mendeley Data, https://doi.org/10.17632/ptz6fzgm7y.1.

The following datasets were generated:

| Author(s) | Year | Dataset title | Dataset URL | Database and Identifier |
|---|---|---|---|---|
| Godfrey W | 2025 | Phosphoglycerate mutase regulates Treg differentiation through control of serine synthesis and one-carbon metabolism | https://www.ncbi.nlm.nih.gov/geo/query/acc.cgi?acc=GSE269846 | NCBI Gene Expression Omnibus, GSE269846 |
| Godfrey W, Kornberg M | 2025 | Godfrey et al_ Phosphoglycerate mutase regulates Treg differentiation through control of serine synthesis and one-carbon metabolism | https://doi.org/10.17632/ptz6fzgm7y.1 | Mendeley Data, 10.17632/ptz6fzgm7y.1 |

The following previously published datasets were used:

| Author(s) | Year | Dataset title | Dataset URL | Database and Identifier |
|---|---|---|---|---|
| Smith KN, Dykema AG, Zhang J | 2023 | Lung tumor-infiltrating Treg have divergent transcriptional profiles and function linked to checkpoint blockade response [human] | https://www.ncbi.nlm.nih.gov/geo/query/acc.cgi?acc=GSE235500 | NCBI Gene Expression Omnibus, GSE235500 |
| Ullah U, Andrabi SBA, Tripathi SK, Dirasantha O | 2016 | Transcriptional repressor HIC1 contributes to suppressive function of human induced regulatory T cells | https://www.ncbi.nlm.nih.gov/geo/query/acc.cgi?acc=GSE90570 | NCBI Gene Expression Omnibus, GSE90570 |
| Schafflick D, Hartlenert M, Schulte-Mecklenbeck A, Lautwein T, Wolbert J, Horste GM | 2019 | Integrated single cell analysis of blood and cerebrospinal fluid leukocytes in multiple sclerosis | https://www.ncbi.nlm.nih.gov/geo/query/acc.cgi?acc=GSE138266 | NCBI Gene Expression Omnibus, GSE138266 |
| Ren S | 2019 | Single cell analysis reveals onset of multiple progression associated transcriptomic remodellings in prostate cancer | https://www.ncbi.nlm.nih.gov/geo/query/acc.cgi?acc=GSE141445 | NCBI Gene Expression Omnibus, GSE141445 |

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
