## [Editor Report · eLife Assessment]

This paper highlights an **important** physiological function of PGAM in the differentiation and suppressive activity of Treg cells by regulating serine synthesis. This role is proposed to intersect with glycolysis and one-carbon metabolism. The study's conclusion is supported by **solid** evidence from in-vitro cellular and in-vivo mouse models.

---

## [Referee Report · Reviewer #1 (Public review)]

Summary:

This work provides a new potential tool to manipulate Tregs function for therapeutic use. It focuses on the role of PGAM in Tregs differentiation and function. The authors, interrogating publicly available transcriptomic and proteomic data of human regulatory T cells and CD4 T cells, state that Tregs express higher levels of PGAM (at both message and protein levels) compared to CD4 T cells. They then inhibit PGAM by using a known inhibitor ECGC and show that this inhibition affects Tregs differentiation. This result was also observed when they used antisense oligonucleotides (ASOs) to knockdown PGAM1.

PGAM1 catalyzes the conversion of 3PG to 2PG in the glycolysis cascade. However, the authors focused their attention on the additional role of 3PG: acting as starting material for the de novo synthesis of serine.

They hypothesized that PGAM1 regulates Tregs differentiation by regulating the levels of 3PG that are available for de novo synthesis of serine, which has a negative impact on Tregs differentiation. Indeed, they tested whether the effect on Tregs differentiation observed by reducing PGAM1 levels was reverted by inhibiting the enzyme that catalyzes the synthesis of serine from 3PG.

The authors continued by testing whether both synthesized and exogenous serine affect Tregs differentiation and continued with in vivo experiments to examine the effects of dietary serine restriction on Tregs function.

In order to understand the mechanism by which serine impacts Tregs function, the authors assessed whether this depends on the contribution of serine to one-carbon metabolism and to DNA methylation.

The authors therefore propose that extracellular serine and serine whose synthesis is regulated by PGAM1 induce methylation of genes Tregs associated, downregulating their expression and overall impacting Tregs differentiation and suppressive functions.

Strengths:

The strength of this paper is the number of approaches taken by the authors to verify their hypothesis. Indeed, by using both pharmacological and genetic tools in in vitro and in vivo systems they identified a potential new metabolic regulation of Tregs differentiation and function.

---

## [Referee Report · Reviewer #2 (Public review)]

Summary:

The authors have tried to determine the regulatory role of Phosphoglycerate mutate (PGAM), an enzyme involved in converting 3-phosphoglycerate to 2-phosphoglycerate in glycolysis, in differentiation and suppressive function of regulatory CD4 T cells through de novo serine synthesis. This is done by contributing one carbon metabolism and eventually epigenetic regulation of Treg differentiation.

Strengths:

The authors have rigorously used inhibitors and antisense RNA to verify the contribution of these pathways in Treg differentiation in-vitro. This has also been verified in an in-vivo murine model of autoimmune colitis. This has further clinical implications in autoimmune disorders and cancer.

[Editors' note: The authors addressed important comments by the reviewers.]

---

## [Author Response]

The following is the authors’ response to the original reviews

**Public Reviews:**

**Reviewer #1 (Public review):**
This work provides a new potential tool to manipulate Tregs function for therapeutic use. It focuses on the role of PGAM in Tregs differentiation and function. The authors, interrogating publicly available transcriptomic and proteomic data of human regulatory T cells and CD4 T cells, state that Tregs express higher levels of PGAM (at both message and protein levels) compared to CD4 T cells. They then inhibit PGAM by using a known inhibitor ECGC and show that this inhibition affects Tregs differentiation. This result was also observed when they used antisense oligonucleotides (ASOs) to knockdown PGAM1.PGAM1 catalyzes the conversion of 3PG to 2PG in the glycolysis cascade. However, the authors focused their attention on the additional role of 3PG: acting as starting material for the de novo synthesis of serine.They hypothesized that PGAM1 regulates Tregs differentiation by regulating the levels of 3PG that are available for de novo synthesis of serine, which has a negative impact on Tregs differentiation. Indeed, they tested whether the effect on Tregs differentiation observed by reducing PGAM1 levels was reverted by inhibiting the enzyme that catalyzes the synthesis of serine from 3PG.The authors continued by testing whether both synthesized and exogenous serine affect Tregs differentiation and continued with in vivo experiments to examine the effects of dietary serine restriction on Tregs function.In order to understand the mechanism by which serine impacts Tregs function, the authors assessed whether this depends on the contribution of serine to one-carbon metabolism and to DNA methylation.The authors therefore propose that extracellular serine and serine whose synthesis is regulated by PGAM1 induce methylation of genes Tregs associated, downregulating their expression and overall impacting Tregs differentiation and suppressive functions.Strengths:The strength of this paper is the number of approaches taken by the authors to verify their hypothesis. Indeed, by using both pharmacological and genetic tools in in vitro and in vivo systems they identified a potential new metabolic regulation of Tregs differentiation and function.

We are grateful to the reviewer for their thoughtful and constructive consideration of our work. We appreciate their comment that the number of approaches taken to test our hypothesis represents a strength that increases confidence in the conclusions.

Weaknesses:Using publicly available transcriptomic and proteomic data of human T cells, the authors claim that both ex vivo and in vitro polarized Tregs express higher levels of PGAM1 protein compared to CD4 T cells (naïve or cultured under Th0 polarizing conditions). The experiments shown in this paper have all been carried out in murine Tregs. Publicly available resources for murine data (ImmGen -RNAseq and ImmPRes - Proteomics) however show that Tregs do not express higher PGAM1 (mRNA and protein) compared to CD4 T cells. It would be good to verify this in the system/condition used in the paper.

This is a fair comment. Although our pharmacologic and genetic studies demonstrated the importance of PGAM in Treg differentiation and suppressive function in murine cells, thereby corroborating the hypothesis formed based on human CD4 cell expression data, we agree that investigating PGAM expression in murine Tregs is important in the context of our work. In reviewing the ImmPres proteomics database, the reviewer is correct that PGAM1 expression was not higher in iTregs compared to other subsets, including Th17 cells. However, when compared to other glycolytic enzymes, expression of PGAM1 increases out of proportion in iTregs. In particular, the ratio of PGAM1 to GAPDH expression is much greater in iTregs compared to Th17 cells. This data is now shown in the revised Figure S5. The disproportionate increase in PGAM1 expression is consistent with the regulatory role of PGAM in the Treg-Th17 axis via modulation of 3PG concentrations, a metabolite that lies between GAPDH and PGAM in the glycolytic pathway. The divergent expression changes between GAPDH and PGAM furthermore support the conclusion that GAPDH and PGAM play opposite roles in Treg differentiation.

It would also be good to assess the levels of both PGAM1 mRNA and protein in Tregs PGAM1 knockdown compared to scramble using different methods e.g. qPCR and western blot. However, due to the high levels of cell death and differentiation variability, that would require cells to be sorted.

We appreciate this comment. As noted by the reviewer, assessing PGAM1 expression via qPCR and Western blot would require cell sorting, which we do not currently have the resources to pursue. However, we measured the effect of ASOs on PGAM1 protein expression using anti-PGAM1 antibody via flow cytometry, which allowed gating on viable cells. As shown in Figure S3A, PGAM-targeted ASOs led to an approximately 40% decrease in PGAM1 expression, as measured by mean fluorescence intensity (MFI). Furthermore, we now show in revised Figure S2 that ASO uptake was near-complete in our cultured CD4 cells.

It is not specified anywhere in the paper whether cells were sorted for bulk experiments. Based on the variability of cell differentiation, it would be good if this was mentioned in the paper as it could help to interpret the data with a different perspective.

Cells were not sorted for bulk experiments. In the revised manuscript, this point is made clear in the text, figure legends, and Methods. It is worth noting that all bulk experiments were conducted on samples with greater than 70% cell viability (greater than 90% for stable isotope tracing studies).

**Reviewer #2 (Public review):**
Summary:The authors have tried to determine the regulatory role of Phosphoglycerate mutate (PGAM), an enzyme involved in converting 3-phosphoglycerate to 2-phosphoglycerate in glycolysis, in differentiation and suppressive function of regulatory CD4 T cells through de novo serine synthesis. This is done by contributing one carbon metabolism and eventually epigenetic regulation of Treg differentiation.Strengths:The authors have rigorously used inhibitors and antisense RNA to verify the contribution of these pathways in Treg differentiation in-vitro. This has also been verified in an in-vivo murine model of autoimmune colitis. This has further clinical implications in autoimmune disorders and cancer.

We very much appreciate these comments about the rigor of the work and its implications.

Weaknesses:The authors have used inhibitors to study pathways involved in Treg differentiation. However, they have not studied the context of overexpression of PGAM, which was the actual reason to pursue this study.

We appreciate this comment and agree that overexpression of PGAM would be an excellent way to complement and further corroborate our findings. Unfortunately, despite attempting several methods, we were unable to consistently induce overexpression of PGAM1 in our primary T cell cultures.

**Recommendations for the authors:**

**Reviewer #1 (Recommendations for the authors):**
I would suggest increasing the font size for flow cytometry gates. Percentages are the focus of the analysis, and it is very hard to read any.

We have increased the font size on all flow cytometry gates, as suggested.

Moreover, most of the flow data show Tregs polarization based on CD25 and FOXP3 expression. However, Figure 3 A, Figure 4D and Figure S3 show Tregs polarization based on FSC and Foxp3. Is there any reason for this?

Antibody staining against CD25 was poor in the experiments noted, which is why Foxp3 alone was used to identify Treg cells in these experiments.

Especially for Figure 3A, other cells could also express Foxp3 making interpretation difficult.

This is a fair comment. With respect to Figures 4D and S3 (now revised Figure S4), these experiments were conducted in isolated CD4 cells, in which the population of CD25-Foxp3+ cells is minimal following Treg polarization (as evident in our other figures). Regarding Figure 3A, previous work has found minimal expression of Foxp3 in circulating non-T cells (Devaud et al., 2014, PMID 25063364), such that we have confidence the identified Foxp3 expressing cells are, in fact, Treg cells. Notably, Figure 3A was already gated on CD4+ T cells, and in the periphery of wild-type mice, these would be reasonably referred to as Tregs, although this does not apply to diseased states or specific cases such as the tumor microenvironment.

The level of murine Tregs differentiation varies a lot among experiments. The % of CD4+CD25+FOXP3+ is ranging from 14% to 77% (controls). It would be good to understand and verify why such differentiation variability.

For most of our Treg polarization experiments, % differentiation in the control group falls within the 35 – 55% range. We found that treatment with ASOs (even scrambled control ASOs) tended to decrease Treg polarization overall, leading to lower numbers of Foxp3 expression in these experiments. Differentiation was similarly low in a few experiments that did not involve the use of ASOs, which we believe was caused by batch variability in the recombinant TGF-b that was used for polarization. Despite this variability, experiments were conducted with sufficient independent experiments and biological replicates to observe consistent trends and to have confidence in the results, as corroborated by statistical testing and the wide variety of experimental approaches used to verify our conclusions. Notably controls were run in every experiment, allowing accurate comparisons to be made in each individual experiment.

Similar comments apply to the level of cell death observed in the cultures of polarizing Tregs.

Although there was some variability in cell viability between experiments, flow cytometry experiments were always gated on live cells, and we believe concerns about reproducibility are substantially mitigated by the number of independent experiments, biological replicates, and distinct experimental approaches used for verification of the experimental findings. For all bulk experiments, cell viability was greater than 70% and equal across samples. For the flux studies, viability was greater than 90% and equal across samples.

Figure 2 B and D: EGCG has been used at two different concentrations. Is it lower in Figure 2D because of one condition being a combination of inhibitors or is it a typo?

The doses stated in the original legend are correct. Yes, drug doses were optimized for combination-treatment experiments. This point is now clarified in the figure legend.

Figure 2G: The description in the results does not match figure legend - Text - serine/glycine-free media or control (serine/glycine-containing) media; figure legend - serine/glycine-free media or media containing 4 mM serine.

We thank the reviewer for pointing out this discrepancy, which was an error in the text. The two conditions used were (1) serine/glycine-free media, and (2) serine/glycine-free media supplemented with 4 mM serine. The text and figure legend have both been updated to clarify this point.

Figure 3 F and G: the graphs do not show the individual points.

Individual points were not shown in these graphs because they are derived from scRNA-seq data, with SCFEA calculated from individual cells. As such, there are far too many data points to display all individual values.

CD4+ T-cell isolation and culture: cells were cultured in 50%RPMI and 50% AIM-V.I thought that AIM-V medium was intended to be for human cultures. Could some of the conditions explain the low level of differentiation observed in some experiments? If there is such variability it might be because the conditions used are not optimal and therefore not reproducible.

We appreciate this critique. Although AIM-V media is often used for ex vivo human T cell cultures, it can similarly be used for mouse T cell culture with the addition of b-mercaptoethanol, as suggested by ThermoFisher and as used in prior publications, such as PMID 36947105. As outlined in the responses above, the differentiation we observed was consistent in most experiments, with some variability based on experimental conditions (such as lower differentiation in the setting of ASO treatment). Furthermore, we believe the number of independent experiments, biological replicates, and independent experimental approaches used in the study supports the reproducibility of our findings.

Figures S1 A, S2 B, and S4: the flow data are shown using both heights (FSC) and area (zombie NIR dye). It would be better to use areas for both parameters.

In the revised manuscript, areas are now used on both the x- and y-axes for these figures.

Figure S1 B and S2 C: The bar graphs are both showing proliferation index, however, the graphs are labelled differently in the two figures and in the legend (proliferation index -Fig S1 B; division index -Fig S2 C and replication index in the legend of Fig S2 C). The explanation of how the index has been calculated should probably go in the legend of the first figure that shows it.

We thank the reviewer for this comment. In the revised manuscript, we have ensured consistency in the terminology (“proliferation index” is now used consistently), and the explanation of the proliferation index calculation is now included in the legend to Figure S1, where the proliferation index first appears.

Were Tregs PGAM1 KD used for RNAseq sorted or not? Based on the plots shown in Figure S2 B there is ~ 50% death which needs to be taken into consideration for the analysis if not depleted.

Similar question for all bulk experiments. It is not specified in the methods or figure legends.

The cells used for RNAseq and other bulk experiments were not sorted. This point is now made clear in the text, figure legends, and Methods. However, cultures were only used for bulk analyses if the viability in those particular experiments was greater than 70%. Given the sensitivity of stable isotope tracing analyses, cultures were only analyzed for those studies if viability was greater than 90%. In these experiments, viability was similar across samples.

It was mentioned in Figure 1 that the PGAM KD led to transcriptional changes that impacted MYC targets and mTORC1 signalling. It would be good to validate these findings maybe with more targeted experiments.

We appreciate this suggestion and agree that validation and further investigation of these critical targets would be worthwhile. However, because of limitations to resources and the fact that these findings are not critical to the main conclusions of the study, we consider these experiments as future directions beyond the scope of the current work.

**Reviewer #2 (Recommendations for the authors):**
Here are a few suggestions and recommendations to improve the research study.(1) The authors have used the word 'vehicle' in most of the figures, however, this word is not explained well in the figure legend. The authors may want to clarify to readers whether vehicle is a plasmid or a solvent for control purposes. For example, in Figure 1D, if vehicle is a plasmid, then another sample for vehicle +/-EGCG should be considered for the rigor in results.

Thank you for identifying this point of confusion. For all drug treatment experiments, vehicle controls consisted of solvent alone without drug. For ASO experiments, the control condition consisted of scrambled ASO. This point is now made clear in the Methods (“Drug and ASO Treatments” section) as well as in the main text. Furthermore, the figure legends and axes have been edited such that “vehicle” is only used to refer to drug experiments (in which solvent vehicle alone was used as control), and “control” is used to refer to ASO experiments (in which scrambled ASO served as control).

(2) Figure 1H represents the RNAseq data for knockdown of PGAM1. It might be interesting to see similar data for the overexpression of PGAM1.

We appreciate this comment and agree that overexpression of PGAM1 would be an excellent way to complement and further corroborate our findings using PGAM1 knockdown and pharmacologic inhibition. Unfortunately, despite attempting several methods, we were unable to consistently induce overexpression of PGAM1 in our primary T cell cultures.

(3) The font in most of the data from flow cytometry experiments (for example 1I) is not legible. Please increase the font size to make it legible.

Font sizes have been increased.

(4) Figure S2, PGAM expression was measured by Flow cytometry experiments. A similar experiment using western Blot, the direct measurement of protein expression, will strengthen the evidence.

We appreciate this comment. As noted in the public reviews, Western blot would require sorting of viable cells, and unfortunately we do not currently have the resources to conduct additional experiments with FACS. However, we respectfully note that assessing protein expression via flow cytometry quantifies protein levels based on antibody binding, similar to Western blot (or in-cell Western blot), while also allowing gating on viable cells. We also note that nearly 100% of cultured CD4 cells took up ASO, as shown in revised Figure S2.

(5) Figure 1J, it is mentioned in the text that 10 datasets were studied. a normalized parameter such as overexpression or suppression could be studied with the variance. It will be good to understand the variability in response among different datasets.

We thank the reviewer for the opportunity to clarify this data. This data was taken from a single published dataset (Dykema et al., 2023, PMID 37713507) in which 10 distinct subsets of tumor-infiltrating Tregs (TIL-Tregs) were identified, rather than from 10 distinct datasets. After identifying the Activated (1)/OX40hiGITRhi cluster of TIL-Tregs as a highly suppressive subset that correlates with resistance to immune checkpoint blockade, Dykema et al. compared gene expression in this subset to the bulked collection of the other 9 subsets, and the data shown in Figure 1J is derived from this analysis. As such, the data in Figure 1J is, indeed, a normalized parameter of overexpression, showing overexpression of PGAM1 in this highly suppressive subset versus other subsets, out of proportion to proximal rate-limiting glycolytic enzymes. The main text and figure/figure legend have been edited to clarify this point.

(6) It will be good to rephrase that the roles of PGAM and GAPDH are opposite, this paragraph is confusing since words such as "supporting Treg differentiation" and "augments Treg differentiation" have been used, although the data in S3 and 1D are opposite. Any possible explanation for the opposing roles of PGAM and GAPDH, despite their involvement in the same pathway of glycolysis, can be added to build up the interest of readers. What is the comparison of the expression of GAPDH and PGAM in Figure 1J?

We thank the reviewer for this comment, as we appreciate that the language used in our initial manuscript was confusing. We have edited the main text, in both the Results and Discussion section, in order to clarify this point and provide explanation as suggested. Indeed, our experimental data indicate that GAPDH and PGAM play opposing roles in Treg differentiation; whereas inhibiting GAPDH activity leads to greater Treg differentiation (shown in revised Figure S4 and our previously published work), similarly inhibiting PGAM leads to diminished Treg differentiation. We view this point (that enzymes within the same glycolytic pathway can have divergent roles in T cells) as a primary implication of these findings, with the explanation that individual enzymes within the same pathway can differentially regulate the concentrations of key immunoactive metabolites. In our study, we identified 3PG as a key immunoactive metabolite whose concentration would be differentially impacted by GAPDH activity versus PGAM activity, since it lies downstream of GAPDH but upstream of PGAM.

To provide further evidence for the opposing roles of GAPDH and PGAM, we analyzed existing datasets. In the revised Figure S5, we show that the PGAM1/GAPDH expression ratio increases in both human and mouse Tregs compared to other CD4 subsets.

(7) Figure 2C, what is M+1, M+2 etc. Does it represent the number of hrs? If so, why are the results for 6 hrs are not shown since the study was for 6 hrs? And what is happening with M+2?

We appreciate the opportunity to clarify this point and apologize for prior confusion. The terminology “M+n” refers to mass-shift produced by incorporation of 13-carbon. When a metabolite incorporates a single 13-carbon atom, it has a mass-shift of one (M+1), whereas incorporation of three 13-carbon atoms produces a mass-shift of three (M+3). Because we used uniformly 13-carbon labeled glucose, 3PG derived from the labeled glucose will have all three carbons labeled (M+3), as will serine that is newly synthesized from 3PG. Because serine can enter the downstream one-carbon cycle and be recycled, we also see the appearance of recycled serine with a single 13-carbon (M+1). The critical point in Figure 2C is that labeled serine is higher in Th17 versus Treg cells, demonstrating that de novo serine synthesis from glycolysis is greater. The main text has been edited to clarify this important point.

(8) Including the quantification of inhibition and rescuing effect of EDCG and NCT will be helpful to readers.

The inhibition and rescuing effects of these drugs are quantified in Figures 2D and 2E as they relate to Treg differentiation. The reviewer may be referring to quantification of relative effects on 3PG levels and serine synthesis. If so, we unfortunately do not have the resources to complete these studies, which would require large-scale quantitative mass spectrometry studies or enzyme activity assays.

(9) Figure 2D and 2E: The authors could also experiment with a dose dependence curve on EGCG and NCT on this phenotype for Treg differentiation. That can help understand the balance between serine pathways and glycolysis pathways. Similarly, the dose dependence of 3PG for Figure 2E and comparing it to the kinetic constants of these enzymes involved and cellular concentrations, these details will be helpful to understand the metabolic dynamics, because this phenotype could be an interplay of both 3PG and serine concentrations.

We appreciate this suggestion and agree that establishing detailed dose-dependence curves and relating these findings to enzyme kinetics would yield additional insights into the biochemical regulation provided by PGAM and PHGDH. Unfortunately we do not have the resources to pursue these additional studies, which therefore lie beyond the scope of our current work.

(10) Figure 4: Explanation for no effect of methionine supplementation?

Thank you for raising this point. We speculate that methionine supplementation had minimal effect because physiologic levels of serine were sufficient to provide basal substrates for the one-carbon cycle. On the other hand, eliminating methionine produced enough of a decrease in one-carbon metabolism to potentiate the effects of excess serine. This point is now briefly addressed in the text.

(11) For direct connection between PGAM and methylation, methylation experiments could be worked out with NCT1 and SHIN1 (as in Figure 4H).

We very much appreciate this suggestion, which we agree would provide a strong complementary approach. Unfortunately we do not have the resources to pursue these studies currently. However, we believe the increased methylation observed following PGAM knockdown (Figure 4G) as strong evidence that PGAM activity directly modulates methylation.